# Spatial and temporal variation in river corridor exchange across a 5th order mountain stream network

Adam S. Ward[1], Steven M. Wondzell[2], Noah M. Schmadel[1,3], Skuyler Herzog[1], Jay P. Zarnetske[4], Viktor Baranov[5,6], Phillip J. Blaen[7,8,9], Nicolai Brekenfeld[7], Rosalie Chu[10], Romain Derelle[11], Jennifer Drummond[7,12], Jan H. Fleckenstein[13,14], Vanessa Garayburu-Caruso[15], Emily Graham[15], David Hannah[7], Ciaran J. Harman[16], Jase Hixson[1], Julia L.A. Knapp[17,18], Stefan Krause[7], Marie J. Kurz[13,19], Jörg Lewandowski[20,21], Angang Li[22], Eugènia Martí[12], Melinda Miller[1], Alexander M. Milner[7], Kerry Neil[1], Luisa Orsini[11], Aaron I. Packman[22], Stephen Plont[4,23], Lupita Renteria[24], Kevin Roche[25], Todd Royer[1], Catalina Segura[26], James Stegen[15], Jason Toyoda[10], Jacqueline Wells[24], Nathan I. Wisnoski[27]

[1] O'Neill School of Public and Environmental Affairs, Indiana University, Bloomington, Indiana, USA

[2] USDA Forest Service, Pacific Northwest Research Station, Corvallis, Oregon, USA.

[3] USGS Earth Surface Processes Division, U.S. Geological Survey, Reston, Virginia, USA

[4] Department of Earth and Environmental Sciences, Michigan State University, East Lansing, Michigan, USA

[5] LMU Munich Biocenter, Department of Biology II, Großhaderner Str. 2, 82152 Planegg-Martinsried, Germany

[6] Department of River Ecology and Conservation, Senckenberg Research Institute and Natural History Museum, 63571 Gelnhausen, Germany

[7] School of Geography, Earth & Environmental Sciences, University of Birmingham, Edgbaston. Birmingham. B15 2TT. UK

[8] Birmingham Institute of Forest Research (BIFoR), University of Birmingham, Edgbaston. Birmingham. B15 2TT. UK

[9] Yorkshire Water, Halifax Road, Bradford, BD6 2SZ

[10] Environmental Molecular Sciences Laboratory, Pacific Northwest National Laboratory, Richland, WA, USA

[11] Environmental Genomics Group, School of Biosciences, the University of Birmingham, Birmingham B15 2TT, UK

[12] Integrative Freshwater Ecology Group, Centre for Advanced Studies of Blanes (CEAB-CSIC), Blanes, Spain

[13] Dept. of Hydrogeology, Helmholtz Center for Environmental Research - UFZ, Permoserstraße 15, 04318 Leipzig, Germany

[14] Bayreuth Center of Ecology and Environmental Research, University of Bayreuth, 95440 Bayreuth, Germany

[15] Earth and Biological Sciences Division, Pacific Northwest National Laboratory, Richland, WA, USA

[16] Department of Environmental Health and Engineering, Johns Hopkins University, Baltimore, Maryland, USA

[17] Department of Environmental Systems Science, ETH Zürich, Zurich, Switzerland

[18] Center for Applied Geoscience, University of Tübingen, Tübingen, Germany

[19] The Academy of Natural Sciences of Drexel University, Philadelphia, Pennsylvania, USA

[20] Leibniz-Institute of Freshwater Ecology and Inland Fisheries, Department Ecohydrology, Müggelseedamm 310, 12587 Berlin, Germany

[21] Humboldt University Berlin, Geography Department, Rudower Chaussee 16, 12489 Berlin, Germany

[22] Department of Civil and Environmental Engineering, Northwestern University, Evanston, Illinois, USA

[23] Department of Biological Sciences, Virginia Polytechnic Institute and State University, Blacksburg, Virginia, USA

[24] Pacific Northwest National Laboratory, Richland, WA, USA

[25] Department of Civil & Environmental Engineering & Earth Sciences, University of Notre Dame, Notre Dame, IN

[26] Forest Engineering, Resources, and Management, Oregon State University Corvallis, OR, USA

[27] Department of Biology, Indiana University, Bloomington, Indiana, USA

*Correspondence to*: Adam S. Ward (adamward@indiana.edu)

**Abstract.** Although most field and modeling studies of river corridor exchange have been conducted a scales ranging from 10's to 100's of meters; results of these studies are used to predict their ecological

and hydrological influences at the scale of river networks. Further complicating prediction, exchanges are expected to vary with hydrologic forcing and the local geomorphic setting. While we desire predictive power, we lack a complete spatiotemporal relationship relating discharge to the variation in geologic setting and hydrologic forcing that are expected across a river basin. Indeed, Wondzell's (2011) conceptual model predicts systematic variation in river corridor exchange as a function of (1) variation in baseflow over time at a fixed location, (2) variation in discharge with location in the river network, and (3) local geomorphic setting. To test this conceptual model we conducted more than 60 solute tracer studies including a synoptic campaign in the 5th order river network of the H.J. Andrews Experimental Forest (Oregon, USA) and replicate-in-time experiments in four watersheds. We interpret the data using a series of metrics describing river corridor exchange and solute transport, testing for consistent direction and magnitude of relationships relating these metrics to discharge and local geomorphic setting. We confirmed systematic decrease in river corridor exchange space through the river networks, from headwaters to the larger mainstem. However, we did not find systematic variation with changes in discharge through time, nor with local geomorphic setting. While interpretation of our results is complicated by problems with the analytical methods, they are sufficiently robust for us to conclude that space-for-time and time-for-space substitutions are not appropriate in our study system. Finally, we suggest two strategies that will improve the interpretability of tracer test results and help the hyporheic community develop robust data sets that will enable comparisons across multiple sites and/or discharge conditions.

## 1 Introduction

Ecological functions and processes in the river corridor are influenced by the exchange of water, solutes, and energy between the surface stream and its catchment, and thus regulate downstream water quality (e.g., Brunke and Gonser, 1997; Krause et al., 2011; Wondzell and Gooseff, 2014; Ward, 2015). These exchange fluxes are collectively termed river corridor exchange and integrate the stream, hyporheic zone, and riparian zone along the river network (Harvey and Gooseff, 2015). Several recent studies have extended feature- and reach-scale findings to predict ecological functions of river corridors at basin scales relevant to resource management (e.g., Gomez-Velez and Harvey, 2014; Kiel and Cardenas, 2014; Gomez-Velez et al., 2015; Bertuzzo et al., 2017; Helton et al., 2018). These approaches require a scaling relationship to predict river corridor exchange across space and through time. Discharge is a logical scaling factor and has been studied as a control on river corridor exchange in both space (i.e., along a network) and time (i.e., under different hydrologic conditions at a fixed location). However, discharge integrates forcing at different scales and may not lead to consistent predictions of river corridor exchange (Ward & Packman, 2018). For example, increases in discharge have been found to cause increases, decreases, or no change in river corridor exchange (Morrice et al., 1997; Butturini and Sabater, 1999; Hart et al., 1999; Jin and Ward, 2005; Wondzell, 2011, 2006; Zarnetske et al., 2007; Schmid, 2008; Karwan and Saiers, 2009; Schmid et al., 2010; Fabian et al., 2011; Ward et al., 2013a). Clearly, to use discharge as a scaling factor to predict river corridor exchange, a more complete description of the exchange-discharge relationship is required.

River corridor exchange is broadly understood to be controlled by interactions between hydrologic forcing and geomorphic setting (Kasahara and Wondzell, 2003; Ward et al., 2012). First, hydrologic forcing encompasses variation in the catchment wetness and storage during storms (Ward et al., 2013a; Dudley-Southern and Binley, 2015; Malzone et al., 2016), seasonal baseflow recession (Payn et al.,

2009; Voltz et al., 2013; Ward et al., 2013c; Schmadel et al., 2017), and diurnal fluctuations arising from natural (e.g., Harman et al., 2016; Musial et al., 2016) or anthropogenic (e.g., Sawyer et al., 2009; Gerecht et al., 2011) controls. While hydrologic forcing reflects a variation in the temporal domain, the geomorphic setting is typically assumed static during river corridor exchange studies. Thus, repeated studies under different discharge conditions are focused on predicting river corridor exchange as a

function of hydrologic forcing and used to develop exchange-discharge relationships at individual study reaches (e.g., Rana et al., 2017). This strategy yields a fixed-in-space, varied-in-time exchange-discharge relationship. Notably, most classical expectations are based on differing steady discharge conditions (e.g., high vs. low baseflow), though an emerging body of field studies (detailed above), modeling studies (e.g., Malzone et al., 2016; Schmadel et al., 2016b), and conceptual models (e.g., Fig.

8 in Ward et al., 2016) are beginning to actively address exchange during unsteady discharge conditions. It is also important to note that, in some cases, changes in discharge can also change the effective geomorphic setting. For example, increases in discharge can flood pool-riffle sequences (e.g., Storey et al., 2003; Church and Zimmerman, 2007) or activate secondary channels (e.g., Ward et al., 2016). Exchange-discharge relationships during steady flow conditions have been examined in many

studies with repeated studies over time at a single site resulting in both positive and negative correlations between river corridor exchange and discharge (Ward and Packman, 2018), though one classic expectation is decreased exchange with increased discharge due to compression of hyporheic flowpaths by toward-stream hydraulic gradients (e.g., Hakenkamp et al., 1993; Hynes, 1983; Palmer, 1993; Vervier et al., 1992; White 1993).

The second primary control on river corridor exchange is the geomorphic setting, including differences attributable to tectonics (e.g., Valett et al., 1996; Payn et al., 2009). Over geologic timescales the geomorphic setting has co-evolved with hydrologic forcing. For example, as drainage area and discharge accumulate through mountain stream networks, we expect predictable spatial patterns

including lower slopes, smaller grain size, larger channel width-to-depth ratios, and increased valley bottom widths (e.g., Leopold and Maddock, 1953; Wohl and Merritt, 2005; 2008; Brardinoni and Hassan, 2007). The evolution of geologic setting occurs over extremely long timescale, allowing the common simplification of assuming geologic setting as static in hyporheic studies. As a result of this assumption, researchers commonly conduct experiments across a spatial gradient to describe patterns in

river corridor exchange (Payn et al., 2009; Covino et al., 2011; Mallard et al., 2014). This approach provides a fixed-in-time, varied-in-space river corridor exchange-discharge relationship that describes a network under a fixed hydrologic condition, most commonly baseflow. Wondzell (1994) suggested that exchange should decrease with increasing watershed size based on first principles. For example, the potential maximum exchange is limited by the streambed area, indicating that the ratio of wetted

perimeter to discharge ($Q$) should be correlated to the maximum possible exchange per unit length of stream channel. As $Q$ increases more rapidly than wetted perimeter as watersheds increase in size, the amount of exchange should be expected to decrease. In fact, most studies have identified a decreasing

role of river corridor exchange as river basins increase in size, attributable to less exchange flux relative to stream flow (Stewart et al., 2011; Mallard et al., 2014; Gomez-Velez and Harvey, 2014; Kiel and Cardenas, 2014; Gomez-Velez et al., 2015; Ward et al., 2018a).

To explain spatiotemporal patterns in river corridor exchange from headwaters to large rivers, Wondzell (2011) developed a conceptual framework describing the relative importance of river corridor exchange to reach-scale transport (i.e., hyporheic exchange flow normalized by river discharge, $Q_{HEF}/Q$), spanning three primary dimensions. First, $Q_{HEF}/Q$ would be largest under the lowest steady-state discharge conditions, where subsurface flow may reflect a larger proportion of total down-valley flow.
Second, $Q_{HEF}/Q$ would be largest in the headwaters and decrease moving toward larger river segments as described above. Lastly, Wondzell (2011) characterized the local geomorphic setting at an individual study site as "hyporheic potential," combining valley slope and hydraulic conductivity to reflect local controls on exchange at the reach scale that might vary locally within the systematic spatial and temporal dimensions. Larger hyporheic potential was associated with larger $Q_{HEF}/Q$. Subsequently,
Harvey et al. (2018) suggested that hydrologic connectivity (i.e., $Q_{HEF}/Q$) is a primary water quality regulator. Ward et al. (2018) further extended this concept to account for changes in valley bottom width and depth of colluvium, describing the down-valley capacity of the valley bottom to transmit water estimated via Darcy's Law. Unlike the first two dimensions, hyporheic potential may not have a predictable trend as one moves down a river continuum, because decreasing slopes and hydraulic
conductivities may be offset by larger hyporheic cross-sections.

Efforts to predict river corridor exchange and associated ecosystem processes as a function of geomorphic setting and hydrologic forcing have been implemented in large-scale remotely sensed test cases. However, this method still lacks field validation across varying discharge and across a range of
stream types with varying morphologic features. For example, Gomez-Velez and Harvey (2014) and Gomez-Velez et al., (2015) used the Networks with Exchange and Subsurface Storage (NEXSS) model to describe spatial patterns in exchange in low-gradient alluvial river networks. NEXSS is based on steady-state discharge and bed sediment grain size as a proxy for local morphologic control. While this modeling approach has demonstrated the importance of river corridor exchange in large river basins, it
is built on scaling relationships derived from idealized mechanistic and conceptual models that may not be representative of headwater streams. Further, the model results have yet to be confirmed in field trials.

To our knowledge, only Payn et al's (2009) field study explicitly considered both spatial and temporal
dimensions of the exchange-discharge relationship. The results of that study were broadly consistent with Wondzell's (2011) conceptual model. However, we now understand that fixed reach lengths cause systematic decreases in the "window-of-Detection" (the timescale of exchange flowpaths that are measurable with tracer studies (Harvey et al., 1996; Wagner and Harvey, 1997; Harvey and Wagner, 2000)). The systematic decrease of window of detection with increasing discharge along the study
stream would have interacted with the fixed reach lengths, likely leading to the underestimation of $Q_{HEF}$ at high discharges. As a result, it is difficult to separate the observed process from limitations of the

measurement instrument (see discussion of Payn et al.'s data in Ward et al., 2013b, and similar studies by Schmadel et al., 2016a).

Several other studies have found general agreement with Wondzell's (2011) prediction of decreasing $Q_{HEF}/Q$ with increasing baseflow through space and at individual study reaches (Kelleher et al., 2013; Patil et al., 2013; Ward et al., 2013c). Thus, Wondzell's (2011) conceptual model might provide an organized framework to extend reach-scale results across space and time in mountain river basins. However, the studies cited above were limited to headwater networks, whereas Wondzell (2011) suggested patterns should hold across much larger scales and geomorphic settings. To date, Wondzell's (2011) conceptual model lacks validation across large river basins studied with a systematic field approach. Given the variability of reach-scale river corridor exchange trends documented in the literature (see summary in Ward and Packman, 2018), it is critical to test Wondzell's (2011) conceptual model with field data that cover much more of the space-time parameter space.

In this study, we seek to characterize river corridor exchange in a mountain stream network as a function of (1) variation in baseflow at a fixed location through seasonal recession; (2) variation in discharge as a function of drainage area during a fixed baseflow condition; and (3) local geomorphic setting (quantified here as hyporheic potential). This study will directly test the conceptual model posed by Wondzell (2011) for mountain stream networks. If the conceptual relationships can be confirmed, this would enable transferability of findings from feature- and reach-scale studies to entire networks of high-gradient mountain streams, paralleling recent advances in low-gradient river networks (e.g., Gomez-Velez and Harvey, 2014; Kiel and Cardenas, 2014; Gomez-Velez et al., 2015). Further, confirmation of the conceptual model would provide a simple scaling relationship for time-variable discharge, which has not been possible to-date. In this study, we conducted a series of solute tracer studies to construct temporal exchange-discharge relationships (i.e., a fixed study reach with observations spanning a range in discharge) and spatial exchange-discharge relationships (i.e., a synoptic campaign to measure exchange at many locations under summer baseflow discharge) for a fifth-order mountain river network, together with physical observations (including hydraulic conductivity, drainage area, slope, valley bottom width, sinuosity) to also characterize hyporheic potential. We interpret the data using a series of metrics describing river corridor exchange and their relationships to discharge.

## 2 Methods

### 2.1 Field site and Solute Tracer Experiments

### 2.1.1 Site Description

The H.J. Andrews Experimental Forest (HJA) is a 5th order basin draining about 6,400 ha in the Western Cascade Mountains, Oregon, U.S.A. with elevations ranging from about 410 to 1,630 m a.m.s.l. The basin is heavily forested and includes stands of old growth Douglas fir trees as well as smaller areas that have been logged to study the effects of forest management practices. Additional

detail about the climate, morphology, geology, and ecology of the site are well described by others (Dyrness, 1969; Swanson and James, 1975; Swanson and Jones, 2002; Jefferson et al., 2004; Cashman et al., 2009; Deligne et al., 2017). The synoptic sampling spanned the entire HJA basin to characterize basin scale valley bottom conditions, while additional more detailed sampling occurred in three distinct landform types.

Headwater sites in the HJA generally fall into one of three landform types associated with underlying geology and geomorphic processes (Table 1). We selected four 2nd order basins to establish fixed stream reaches for replication through the summer baseflow recession period, one in each landform type plus one replicate. The first landform type occurs in the lower elevations of the HJA where geology is dominated by Upper Oligocene - Lower Miocene basaltic flows. These volcanoclastic rocks were weakened by hydro-thermal alteration from subsequent volcanic activity, enabling rapid downcutting and formation of a highly dissected landscape. Hillslopes are steep; valleys are v-shaped and tend to be narrow with steep longitudinal gradients. Valley bottom colluvium is typically shallow but variable, being emplaced by hillslope mass wasting and debris flows. Exposed bedrock is visible in many locations, while deeper deposits form behind individual large logs or larger log jams. We selected the well-studied Watersheds 1 and 3 (WS01 and WS03) for two of our fixed reaches (Figure 1). Briefly, WS01 and WS03 valley bottoms reflect different time periods in this landform. In 1996, WS03 was scoured to bedrock along 100s of meters of the valley bottom (Johnson, 2004). Since that time no debris flows have been recorded, resulting in a study reach nearly free of colluvium in the upper half of the study reach. WS01 is a paired catchment to WS03, reasonably representing a pre-scour and less-constrained comparison to WS03. WS01 has a wood-forced step-pool morphology (Montgomery and Buffington, 1997; 1998) over most of its mainstem length, representative of many steep mountain streams. River corridor exchange in the two catchments have been broadly studied using a paired catchment approach (e.g., Wondzell, 2006; Voltz et al., 2013; Ward et al., 2017b).

Deep-seated earth flows provide a second contrasting landform type in the HJA. These are emplaced on the Upper Oligocene - Lower Miocene basaltic flows and are characterized by a poorly developed channel network (many parallel channels), a general lack of lateral contributing area to the river corridor, little lateral constraint, and extensive colluvial deposits with no bedrock exposure. Based on visual inspection, channels on these earthflows are actively meandering, braiding, and downcutting. Characteristic geomorphic features include meander bends and cut-banks (visually similar to lower-gradient alluvial systems of the region) in addition to step-pool features. We selected an unnamed 2nd order reach on a large earth flow adjacent to WS03 for this study (Figure 1).

The third landform type occurs in high elevation headwater catchments with U-shaped valleys characteristic of glacial cirques, which formed in plieocascade volcanics. Valley bottoms are filled with compacted glacial tills. Large wood atop the till forms pools and steps with intermediate gravel and cobble riffles. Lateral tributary area is relatively uniform along the valley with few hollows or tributary valleys (in contrast to the highly dissected landforms in WS01 and WS03). Bedrock is rarely visible along the study site. We selected a 2nd order reach of Cold Creek to represent this landform (Figure 1).

**Table 1. Summary of key characteristics of the fixed-reach sites. See detailed descriptions for further information (*Dyrness*, 1969; *Swanson and James*, 1975; *Swanson and Jones*, 2002; *Jefferson et al.*, 2004; *Cashman et al.*, 2009; *Deligne et al.*, 2017)**

| Site | Important Hydrologic Controls | Important Geologic Controls |
|------|-------------------------------|------------------------------|
| WS01 | <ul><li>Highly-dissected landscape</li><li>Focused lateral inflows</li><li>Diurnal discharge fluctuations due to evapotranspiration</li></ul> | <ul><li>Colluvium deposited by debris flows from hillslopes forms extensive deposits in the valley bottom</li><li>V-shaped, rapidly downcutting valley</li></ul> |
| WS03 | <ul><li>Highly-dissected landscape</li><li>Focused lateral inflows</li><li>Diurnal discharge fluctuations due to evapotranspiration</li></ul> | <ul><li>Scoured to bedrock in 1996 leaving only small colluvial deposits</li><li>Highly constrained, low colluvium analogue to WS01</li><li>V-shaped, rapidly downcutting valley</li></ul> |
| Unnamed Cr. | <ul><li>Surficial aquifer on earthflow connects several parallel channels</li><li>Minimal lateral tributary area</li></ul> | <ul><li>Deep-seated earthflow</li><li>No defined valley; parallel stream channels down hillslope</li></ul> |
| Cold Cr. | <ul><li>Extensive aquifer provides high discharge, cold baseflow year-round</li><li>Diffuse lateral inflows</li></ul> | <ul><li>Compressed glacial tills</li><li>U-shaped valley (glacial cirque)</li><li>Uniform lateral tributary area</li></ul> |

**2.1.2 Synoptic study**

We conducted a synoptic study at 46 sites within the HJA during late summer baseflow conditions (Figure 1) that included solute tracer experiments. Site selection was stratified by by stream order so that more headwater sites were sampled than higher order reaches, as suggested by other synoptic investigations of sediment-water interfaces at the basin scale (Ruhala et al., 2017; Lee-Cullin et al.,

2018). We selected low baseflow conditions to maximize our ability to measure $Q_{HEF}/Q$, which is expected to be largest under low discharge conditions (Wondzell, 2011). Study sites were selected to achieve coverage across stream orders, landforms, and on the basis of accessibility from roads in the basin. The data described here are documented and field methods described in detail by Ward et al. (2019), but we provide an overview below.

At each site we measured mean stream width and depth, valley width, and collected GPS coordinates. Subsequently, a modified version of TopoToolbox 2.0 (Schwanghart and Kuhn, 2010; Schwanghart and Scherler, 2014) and a 1-m LiDAR derived, digital elevation model (DEM) was used to extract upslope accumulated area (UAA; ha), valley slope ($S_{val}$; m m$^{-1}$), and a stream centerline that was used to

calculate sinuosity (Sinuosity; m/m) . Our methods were identical to those previously used in the basin (Corson-Rikert et al., 2016; Schmadel et al., 2017; Ward et al., 2018c, 2018a).

At each synoptic site, we drove a Solinst 615N well point into the streambed so that the top of the 0.15 m screened interval was 50-cm below the streambed. After developing the well with a peristaltic pump,

we conducted 3 to 6 replicate falling head tests, measuring head change through time using a down-well

Van Essen Micro-Diver logging at a 0.5-s interval. Falling head tests were interpreted using the Hvorslev (1951) method:

$$K = \frac{r^2 \ln\left(\frac{L_e}{R}\right)}{2 L_e T_0}$$

where $K$ is hydraulic conductivity (m/s), $r$ is the radius of the well casing (0.025 m), $R$ is the radius of the well screen (0.005 m), $L_e$ is the screened length of the well (m), and $T_0$ is the time for the head to fall to about 37% of its original value (i.e., the e-folding time; s). We took the geometric mean of the replicate tests as the representative value of $K$ at each site.

We calculated the capacity of the subsurface to convey water down the valley bottom ($Q_{sub,cap}$, sometimes termed "underflow"; m$^3$ s$^{-1}$) as:

$$Q_{sub,cap} = b_{valley} h_{valley} K S_{val}$$

following *Ward et al.* (2018a), where $b_{valley}$ is the valley width, $h_{valley}$ is the valley colluvium depth (m; estimated as 50% of the wetted channel width. This estimate is consistent with depths used in past studies (Gooseff et al., 2006; Ward et al., 2012; Crook et al., 2008; Ward et al., 2018a; 2018c; Schmadel et al., 2017 ) and geophysical transects in the 4th and 5th order reaches of Lookout Creek (Wondzell, unpublished data). We calculated hyporheic potential ($HYP_{POT}$; m s$^{-1}$) after Wondzell (2011), a similar metric that does not account for valley width, depth, or porosity, as:

$$HYP_{POT} = S_{val} K$$

We also calculated stream power ($\Omega$; W m$^{-2}$) at each tracer release location as:

$$\Omega = \rho g Q S$$

where $\rho$ is the density of water (kg m$^{-3}$), $g$ is the gravitational constant (9.81 m s$^{-2}$), $Q$ is the average discharge in the study segment (m$^3$ s$^{-1}$), and $S$ is the DEM-derived slope along the stream channel in the study segment (m m$^{-1}$).

Finally, at each site, we established a stream-tracer study reach with length approximately 20 times the wetted channel width that would be representative of reach-scale morphologic variation (MacDonald et al., 1991; Montgomery and Buffington, 1997; Rot et al., 2000; Martin, 2001; Anderson et al., 2005). We instantaneously released a known mass of NaCl (assumed conservative), dissolved in stream water, one mixing length (i.e., the distance required for the solute tracer to be well-mixed across the channel cross-section) from the downstream end of the study reach, where we monitored in-stream specific conductance (Onset Computer Corporation, Bourne, MA, USA). Mixing lengths were based on visual estimates in the field as empirical estimates are unreliable in mountain streams (Day et al., 1977).

Moreover, field experience in a study system is recognized to be potentially more useful that theoretical estimates of mixing length (Kilbatrick and Cobb, 1985). Thus, we used visual estimates that are consistent with our past studies using these techniques and tracers in H.J. Andrews Experimental Forest (Ward et al., 2012; 2013a; 2013b; 2019; Voltz et al., 2013) and practices used in other mountain stream

networks (e.g., Payn et al., 2009; Covino et al., 2010). Next, we released a second known mass of NaCl one mixing length above the upstream end of the study reach. We monitored in-stream specific conductance at both the up- and downstream ends of the study reach. Mixing lengths were visually estimated in the field; small amounts of a fluorescent dye were used to assess mixing lengths where they could not be readily determined by surface hydraulic conditions. All in-stream specific conductance

measurements were converted to concentrations of NaCl mass added using a 4-point calibration curve developed from standards made by mixing varying amounts of NaCl with stream water that encompassed the range of observations during the tracer tests. Results from all sensors were composited into a single linear regression ($r^2 > 0.99$).

### 2.1.3 Fixed-reach studies

We established 11 fixed reaches of about 50-m of valley length in the four headwater catchments. We conducted identical site characterizations as described above for the synoptic study. However, for each study reach, solute tracer injections were conducted 2-6 times through baseflow recession. The differing number of replicates reflects either sensor failure or omission of a replicate due to conflicting research occurring at the same sites by other researchers (i.e., our replication would have negatively impacted

their independent research campaigns, so we did not proceed with our injections). These sites parallel the common approach of replication of a study at a fixed reach with varied discharge to relate river corridor exchange to discharge conditions (after Payn et al., 2009).

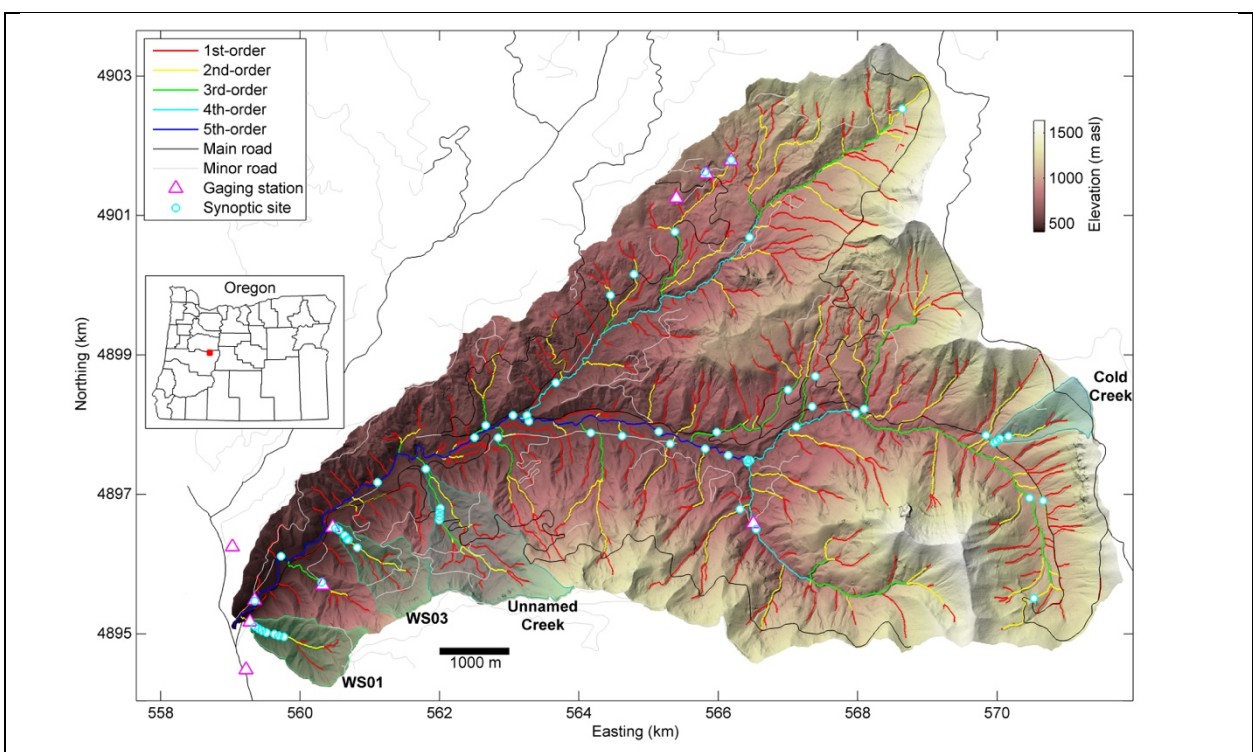

**Fig. 1. Synoptic study sites and LiDAR-derived stream network for the H.J. Andrews Experimental Forest. Reprinted with permission from *Ward et al.* (2019).**

### 2.1.4 Reach length and study design

In the synoptic campaign, we scaled our tracer reach lengths by wetted channel width in an effort to control for the advective timescales of the study. To demonstrate how this decision, or conversely the decision to fix our study reach in headwaters, may have biased our data collected, we conducted a series of four tracer injections in first through fourth stream orders in the study basin. For each study we fixed a single location for the injection and placed sensors downstream at three distances: (1) a fixed reach of 150 m; (2) an estimated 10-min of advective time downstream, based on timing debris floating along approximately 5-m of stream; and (3) a distance of 20 times the wetted channel width, which was identified as a length scale for a representative study reach in the HJA (Anderson et al., 2005; Gooseff et al., 2006). All injection protocols were consistent with synoptic and replicate injections described above.

### 2.2 Analysis of stream solute tracer injections

There exists no single, widely agreed upon, robust framework for describing river corridor exchange based on stream solute tracer experiments. Instead, a host of approaches have been successfully used to interpret experimental data. In this section we detail the interpretation of stream solute tracers using

several established approaches. Notably, the interpretations here were selected because they most directly interpret the observed solute tracer timeseries, in contrast to other strategies that focus on inverse model parameterization (e.g., Bencala and Walters, 1983; Haggerty and Reeves, 2002) and may be prone to parameter uncertainty and identifiability challenges (e.g., Ward et al., 2017a; Kelleher et al., 2013; Rana et al., 2019; Rana et al., 2019). The suite of approaches implemented here were selected because the provide complimentary interpretations that may be informative when jointly considered (Table 2). We emphasize here that we do not seek a singular, "best" metric to describe river corridor exchange, but instead seek to interpret a suite of metrics to provide a comprehensive understanding of our study system.

**Table 2. Summary of solute tracer interpretation strategies**

| Approach | Strengths | Limitations | Key Metrics | Method Documentation Example Studie |
|---|---|---|---|---|
| Separation of advection-dispersion from transient storage | • Quantifies the relative importance of 1-D advection-dispersion in the stream channel in comparison to transient storage | • Only considers recovered mass; no interpretation of lost mass.<br>• Enforces separation of in-stream advection-dispersion from other processes.<br>• Relies upon idealized model of advection-dispersion. | • Fraction of recovered mass primarily involved in advection-dispersion ($f_{MAD}$)<br>• Fraction of recovered mass primarily involved in transient storage ($f_{MTS}$) | Wlostowski et a (2017); Ward et (2018b); Harms al. (2019) |
| Short-term storage analysis | • Summary metrics for the time-integrated, recovered mass.<br>• Relates timescales of storage to length-scales of subsurface flowpaths | • Only considers recovered mass; no interpretation of lost mass.<br>• Assumes steady discharge (as formulated in this study, but can be modified)<br>• Sensitive to late-time noise and errors in mass recovery | • Median arrival time ($M_1$)<br>• Coefficient of variation ($CV$)<br>• Skewness ($\gamma$)<br>• Holdback ($H$)<br>• Longest detectable subsurface flowpath ($L_{detect}$)<br>• Time at which 99% of recovered mass is recovered ($t_{99}$)<br>• Transient storage index ($TSI$, not presented in this study. $TSI = t_{99} - t_{peak}$) | Gupta and Cvetkovic (200( Gooseff et al. (2007); Ward et (2010b); Schma et al. (2016) |
| StorAge Selection (SAS) analysis | • Relates outflow timeseries to age of water.<br>• Does not require mechanistic (e.g., advection-dispersion) formulation.<br>• No artificial division into stream and storage domains<br>• Estimates volume of storage sampled by tracer | • Only considers recovered mass; no interpretation of lost mass.<br>• Metrics do not map directly to classical transport mechanisms | • Storage volume sampled by tracer ($S_{comp}(T)$)<br>• Discharge sampled by tracer ($Q_{comp}(T)$)<br>• Fraction of storage volume sampled ($f_{VTOT}(T)$)<br>• Fraction of discharge sampled ($fQ_{labelled}(T)$)<br>• Fraction of stream volume sampled ($f_{VSTR}$) | Harman (2015); Harman et al. (2016) |
| Long-term storage analysis | • Characterizes the fate of mass beyond the window of detection | • No interpretation of recovered tracer mass<br>• Bounds a plausible range of gross gains and losses | • Maximum gross losses of stream water from the study reach ($Q_{LOSS,MAX}$)<br>• Maximum gross gains of stream water to the study reach ($Q_{GAIN,MAX}$)<br>• Maximum gross losses of stream water from the study reach ($Q_{LOSS,MIN}$, not presented in this study)<br>• Maximum gross gains of stream water to the study reach ($Q_{GAIN,MIN}$, not presented in this study) | Payn et al. (200( Covino et al. (2011); Payn et (2012); Mallard al. (2014) |

### 2.2.1 Separation of advection-dispersion from transient storage

We separated the recovered solute tracer mass into fractions that were primarily related to advection-dispersion and to short-term transient storage (after Wlostowski et al., 2017). Briefly, stream velocity ($v$; m s$^{-1}$) is estimated as $v = L/t_{peak}$, where $L$ is the length of the reach along the centerline, and $t_{peak}$ is
the time at which the peak breakthrough curve concentration is observed, interpreted as the advective timescale of the study reach. The stream cross-sectional area ($A$; m$^2$) is estimated by $A = Q_{DS}/v$, where $Q_{DS}$ is an estimate of discharge at the downstream end of the study reach based on dilution gauging. The mass of solute tracer recovered from the upstream injection at the downstream end of the study reach ($M_{REC}$; g) is calculated as:

$$M_{REC} = Q_{DS} \int_0^{t_{99}} C_{obs}(t) dt$$

where $C_{obs}$ (g m$^{-3}$) is the observed solute tracer concentration at the downstream location in response to the upstream solute tracer injection. Using these estimates, the analytical solution to the advection-
dispersion equation given the instantaneous tracer addition method is:

$$C_{ADE}(t) = \frac{M_{REC}}{A(4\pi Dt)^{1/2}} exp\left[\frac{(L - vt)^2}{4Dt}\right]$$

where $C_{ADE}$ (g m$^{-3}$) is the concentration time-series predicted for the recovered mass transported via
advection and dispersion only, $M_{REC}$ is mass recovered (g), and $D$ is the best-fit longitudinal dispersion coefficient (m$^2$ s$^{-1}$). Following this approach, the concentration time-series for a solute that is predominantly transported by advection and dispersion ($C_{AD}$) can be estimated as:

$$C_{AD}(t) = \begin{cases} C_{obs}(t); & C_{ADE}(t) > C_{obs}(t) \\ C_{ADE}(t); & C_{ADE}(t) < C_{obs}(t) \end{cases}$$

The total mass associated with advection and dispersion ($M_{AD}$) can be calculated as:

$$M_{AD} = \int_0^{t_{99}} C_{AD}(t) Q_{DS} dt$$

where $t_{99}$ (s) is the time at which 99% of the recovered tracer signal has passed by the monitoring location. The component of $C_{obs}$ that is primarily impacted by transient storage ($C_{TS}$, g m$^{-3}$) can be calculated as:

$$C_{TS} = C_{obs} - C_{AD}$$

Similar to $M_{AD}$, the mass associated with transient storage ($M_{TS}$, g) can be calculated as:

$$M_{TS} = \int_0^{t_{99}} C_{TS}(t)Q_{DS}dt$$

5 Finally, we calculate the fraction of recovered mass primarily involved in advection-dispersion ($f_{MAD}$) or transient storage ($f_{MTS}$) as:

$$f_{MAD} = \frac{M_{AD}}{M_{REC}}$$

and

$$f_{MTS} = \frac{M_{TS}}{M_{REC}}$$

### 2.2.2 Short-term storage analysis

Observations of stream solute tracer releases were analyzed using a host of time-series metrics. We calculated the time at which 99% of the total mass recovery was achieved ($t_{99}$; s). To minimize the
15 impacts of late-time noise on calculated metrics, $C_{obs}$ was truncated at the downstream end to only include times bounded by the injection time and $t_{99}$ (hereafter $C_{obs}(t)$), consistent with common practices (e.g., Mason et al., 2012; Ward et al., 2013a; 2013b; Schmadel et al., 2016a) and a community tool for interpretation of solute tracers (Ward et al., 2017a). The truncated time-series was normalized to isolate the features of the data in the temporal domain and minimize effects of different concentration
20 magnitudes between injections. The normalized breakthrough curve ($c(t)$) was calculated as:

$$c(t) = \frac{C_{obs}(t)}{\int_{t=0}^{t_{99}} C_{obs}(t)dt}$$

We calculated the median arrival time ($M_1$; equivalent to the first temporal moment; s) as:

$$M_1 = \int_{t=0}^{t_{99}} tc(t)dt$$

Next, we calculated the 2nd and 3rd order moments about $M_1$ ($\mu_2$ and $\mu_3$) as:

$$\mu_n = \int_{t=0}^{t_{99}} (t - M_1)^n c(t)dt$$

where $n$ represents the $n^{th}$ order moment, and $\mu_2$ and $\mu_3$ contain information about symmetrical and asymmetrical spreading of the time-series, respectively. The central moments were normalized to provide information that could be compared between sites and injections by calculating the coefficient of variation ($CV$) and skewness ($\gamma$) as:

$$CV = \frac{\mu_2^{1/2}}{M_1}$$

$$\gamma = \frac{\mu_3}{\mu_2^{3/2}}$$

Finally, we calculated the holdback of the system ($H$), which describes transport in a continuum ranging from piston flow ($H = 0$) to no movement of the solute ($H = 1$) (Danckwerts, 1953). Ward et al. (2018b) interpret higher values of $H$ to indicate greater influence of transient storage on reach-scale transport. Holdback is calculated as:

$$H = \frac{1}{M_1} \int_{t=0}^{M_1} F(t)dt$$

where

$$F(t) = \int_{\tau=0}^{t} c(\tau)d\tau$$

Finally, we estimated the maximum detectable flowpath length ($L_{detect}$) as:

$$L_{detect} = t_{99}\frac{K}{\theta}S_{val}$$

which is based on Darcy's Law, but uses the valley slope ($S_{val}$) as an estimate of the hydraulic gradient (after Wondzell, 2011; Ward et al., 2017b) and where $\theta$ is porosity.

### 2.2.3 StorAge Selection (SAS) analysis

We interpreted the transport of tracer through the study reach using the StorAge Selection (SAS) approach (Harman, 2015; Harman et al., 2016). Briefly, this approach can be used to describe the composition of outflowing water from a study reach as a combination of water sampled from different

ages within the study reach. The approach is closely related to transit time distributions, but isolates the contribution to the transit time of storage turnover from that of inflow and outflow variability. Although physically-based, in the sense of conforming to conservation of mass and describing physically meaningful properties, this approach describes the higher-level emergent effects of mechanisms like advection, dispersion and other processes (Harman et al., 2016). Instead, the approach provides a description of the reach as a zero-dimensional, integrated control volume (i.e., no arbitrary division of surface vs. subsurface or mobile vs. less mobile storage).

Here, we closely follow the adaptation of the general formulation of the SAS framework to interpret stream solute tracer results (Harman et al., 2016). Notably, we are able to further simplify the approach by assuming discharge was steady-state during each injection and having only a single release of tracer that did not overlap with other tracer signals. Under the assumption of steady flow the forward and backward transit time distributions are equal. First, we calculated the probability density of the (forward) transit time distribution ($p_Q(T)$) as:

$$p_Q(T) = \frac{QC_{obs}}{M_{US}}$$

where $M_{US}$ is the mass of the upstream tracer injection (g). Note that, due to the steady state assumption, $p_Q(T)$ is only a function of water age $T$ and does not depend on time $t$. Next, we calculated the cumulative form of the transit time distribution ($P_Q(T)$) as:

$$P_Q(T) = \int_{\tau=0}^{T} p_Q(\tau)d\tau$$

where $\tau$ is a random variable representing the age of a parcel of water (Harman, 2015). This allows us to determine the age-rank discharge ($Q_T(T)$):

$$Q_T(T) = QP_Q(T)$$

and the age-ranked storage ($S_T(T)$) as:

$$S_T(T) = Q\left(T - \int_{\tau=0}^{T} P_Q(\tau)d\tau\right)$$

The age-rank storage can be interpreted to determine the volume of reach storage that was sensed by the tracer. If the total storage in the study reach can be estimated, the fraction of total storage that was sensed by the tracer can also be determined. A perfect tracer study would be sensitive to the entirety of the storage volume. However, due to limitations arising from the window of detection and truncation of

the breakthrough curve, only a fraction of the storage is actually measured (e.g., Drummond et al.*,* 2012). The knowledge of measured volume is important and is one advance enabled by using this interpretation framework.

Plotting the age-rank discharge as a function of the corresponding age-rank storage reveals the SAS function (Harman 2015; Harman et al. 2016). This relationship shows how discharge is composed of water drawn from storage of different ages. Flipping this plot along each axis to plot the complements is advantageous to interpret the results (Harman et al., 2016). Thus, we plot the age rank discharge complement

$$Q_{comp}(T) = Q(1 - P_Q(T))$$

as a function of the age-rank storage complement

$$S_{comp}(T) = S_{ref} - S_T(T)$$

where $S_{ref}$ is the total storage in the study reach (m³). We estimated $S_{ref}$ as the volume of the surface water (mean width × mean depth × length along centerline) plus the subsurface storage volume (valley width × valley segment length × depth × porosity). We estimated porosity as 30% for all locations (after
Domenico and Schwartz, 1990; Ward et al., 2018a).

The SAS analysis can be interpreted to yield an understanding of how storage and discharge are related for the study. The minimum value of the age-rank discharge complement (Y-axis of Fig. 2) gives the discharge of outflowing water in the channel that was not labeled by the tracer at the upstream end of
the study reach within the window of detection. In practice, unlabeled discharge represents some combination of (1) down-valley flow entering the segment from upstream and then upwelling, and (2) discharge originating from parts of storage that retain tracer for very long periods of time. Finally, while both the discharge and volume sampled will scale through the network, each can be normalized to a reference value as:

$$f_{VTOT}(T) = \frac{S_{comp}(T)}{S_{ref}}$$

$$f_{Q,labeled}(T) = \frac{Q_{comp}(T)}{Q + Q_{sub,cap}}$$

where $f_{VTOT}$ is the fraction of the total storage volume that was sampled with the tracer and $f_{Q,labeled}$ is the fraction of the total down-valley discharge that was labeled with the tracer. We also calculated the fraction of the in-stream volume sampled ($f_{VSTR}$) as:

$$f_{VSTR} = \frac{S_{comp}(T)}{AL}$$

The SAS approach requires a physically plausible bounding by input values. In practice, this means that
5   errors in discharge can cause overestimations of mass recovery (i.e., greater than the mass that was
injected), leading to physically impossible $Q_T(T)$ values. As a result, we assumed a typical error of 10%
for dilution gauging (Schmadel et al., 2010). Within that range of discharge values, we calculated the
range of physically plausible discharges (i.e., those which yield physically meaningful SAS
calculations), and analyzed the midpoint of the plausible range. In the first study using the SAS
10   approach to interpret solute tracers, Harman et al., (2016) found that a similar discharge adjustment was
required to define the feasible parameter space.

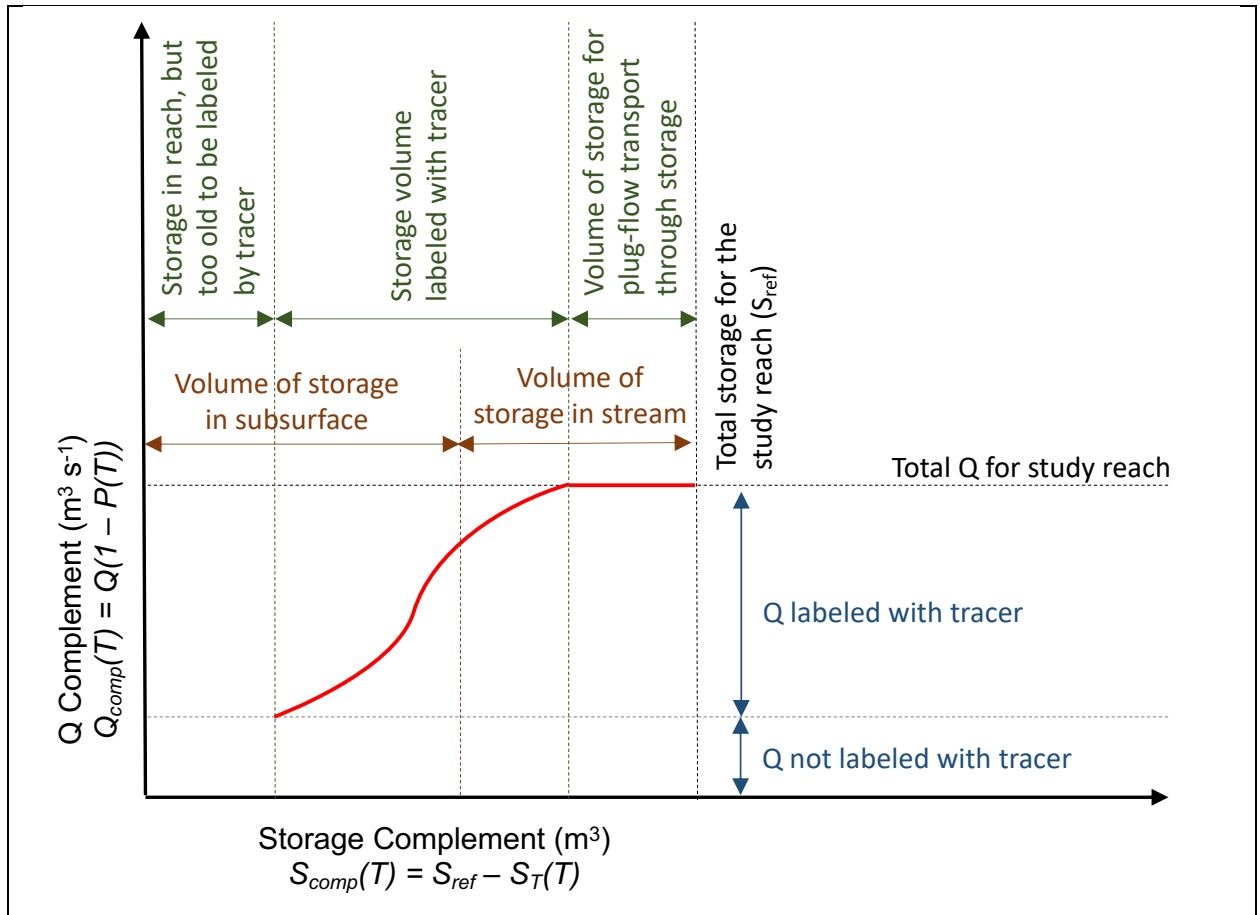

**Fig. 2. Graphical representation and interpretation the SAS function. Note that the volume of storage in the stream vs. subsurface (orange above) is independent of the SAS analysis and is provided here as an example of integrating the SAS metrics with other knowledge about the system.**

**2.2.4 Long-term storage analysis**

Long-term storage characterized the fate of mass beyond the window of detection (i.e., unrecovered mass that did not contribute to the analysis of short-term storage; Payn et al., 2009; Ward et al., 2013c). Dilution gauging at the up- and downstream ends of each study reach was used to estimate discharge ($Q_{US}$ and $Q_{DS}$, respectively; m$^3$ s$^{-1}$). Mass loss along the study reach can be calculated by the difference of the mass injected ($M_{US}$; g) and $M_{REC}$:

$$M_{LOSS} = M_{US} - M_{REC}$$

Finally, Payn et al. (2009) demonstrate how $M_{LOSS}$, $Q_{US}$, and $Q_{DS}$ can be used to bound the gross gains and losses of water to the channel through the study reach. We focus here on the case of all losses occurring before all gains, which is the end-member that yields the largest estimates for gross losses ($Q_{LOSS,MAX}$) and gains ($Q_{GAIN,MAX}$) respectively, calculated as:

$$Q_{LOSS,MAX} = \frac{M_{LOSS}}{\int_0^{t_{99}} C_{obs,ds}(t)dt}$$

$$Q_{GAIN,MAX} = Q_{DS} - Q_{US} - Q_{LOSS,MAX}$$

The net change in discharge along the study reach ($\Delta Q$) is represented by the terms $Q_{DS}$-$Q_{US}$ in the equation above. To compare between reaches, we normalized $M_{LOSS}$ by $M_{INJ}$ and normalized the gross gains and losses by $Q_{US}$. We also calculate gross gains and gross losses, $f_{QGAIN,MAX}$ and $f_{QLOSS,MAX}$, as a fraction of the inflow at the upstream end of the reach.

**2.3 Statistical Tests**

We applied a Mann-Kendall (MK) test to examine relationships between the metrics of river corridor exchange and characteristics of geologic setting and hydrologic forcing. The MK test is a non-parametric test used to assess the likelihood of a monotonically increasing or decreasing trend in a data set, which we interpret as the presence of a systematic trend through the river network. The MK test only provides an indication of a relationship's existence and does not characterize the direction nor magnitude of the relationship. Thus, we also calculated Sen's slope, a non-parametric test to fit a robust linear slope to a data set by choosing the median of slopes connecting all potential pairs of points. This metric was selected because it is less sensitive to outliers than a traditional linear regression and more robust for skewed or heteroskedastic data. Thus, we use the MK test to define the presence or absence of a statically significant trend (p<0.05) and Sen's slope to indicate the direction of that trend (positive

or negative). We also compare the magnitude of Sen's slope among and within datasets to estimate the relative sensitivity of selected dependent variables to the same independent variable.

For the synoptic data we also report the coefficient of determination ($r^2$) for univariate best-fit power-law regression as an indicator of the predictive power of a parsimonious model fit. The coefficient of determination is commonly interpreted as the percent of variance explained by the model. We selected a power law regression because most independent and dependent variables span orders of magnitude. We did not test other functional forms as the purpose of this fit is to assess the explanatory power of a simple regression-model -- comparable to those commonly used to interpret field data for identifying relationships between two variables -- rather than identify an optimal predictive equation that relates the two variables. Finally, we fit a planar surface to each metric as a function of log-transformed baseflow and $HYP_{POT}$ to approximate the conceptual model proposed by Wondzell (2011). We selected a planar surface in log-space as the simplest representation of a relationship. We also fit univariate linear relationships to the log-transformed $Q$ and $HYP_{POT}$ data for each metric. We emphasize here our foucs was on attesting Wondzell's (2011) conceptual model, not an exhaustive curve- nor surface-fitting exercise.

## 3 Results

### 3.1 Spatial patterns in hydrologic and geomorphic controls

Overall, all landscape metrics exhibited statistically significant monotonic trends with one another (MK test; $p < 0.05$). We found expected trends of increasing $UAA$ (Fig. 3A) velocity (Fig. 3B) and stream order (Fig. 3C) with discharge. We also found an increasing hydraulic conductivity in the down-network direction (Fig. 3D), which is indicative of sediment size and sorting in high-relief headwater landscapes (Brummer and Montgomery, 2003), but opposite to typical low-relief alluvial systems (e.g., Gomez-Velez et al., 2015). Moving from the headwaters to the outlet, we found flattening and widening of the valley with increasing discharge and $UAA$ along the network (Figs. 3E, 3F), increasing stream power (Fig. 3G), and increasing sinuosity (Fig. 3I),  This trend reflects the prevalence of fine material in the upper reaches emplaced by debris flows and coarsening in the downstream direction where stream power increases thus exporting fines from the system. The result of these trends in valley morphology and hydraulic conductivity is an increasing trend in $Q_{sub,cap}$ in lower network positions (Fig. 3H), indicating the increasing width and $K$ are sufficient to overcome the decreases in slope in generating this relationship. Pairwise Pearson correlation coefficients and Spearman Rank correlation coefficients are summarized in supplemental figures S3 and S4, and table S1 and S2.

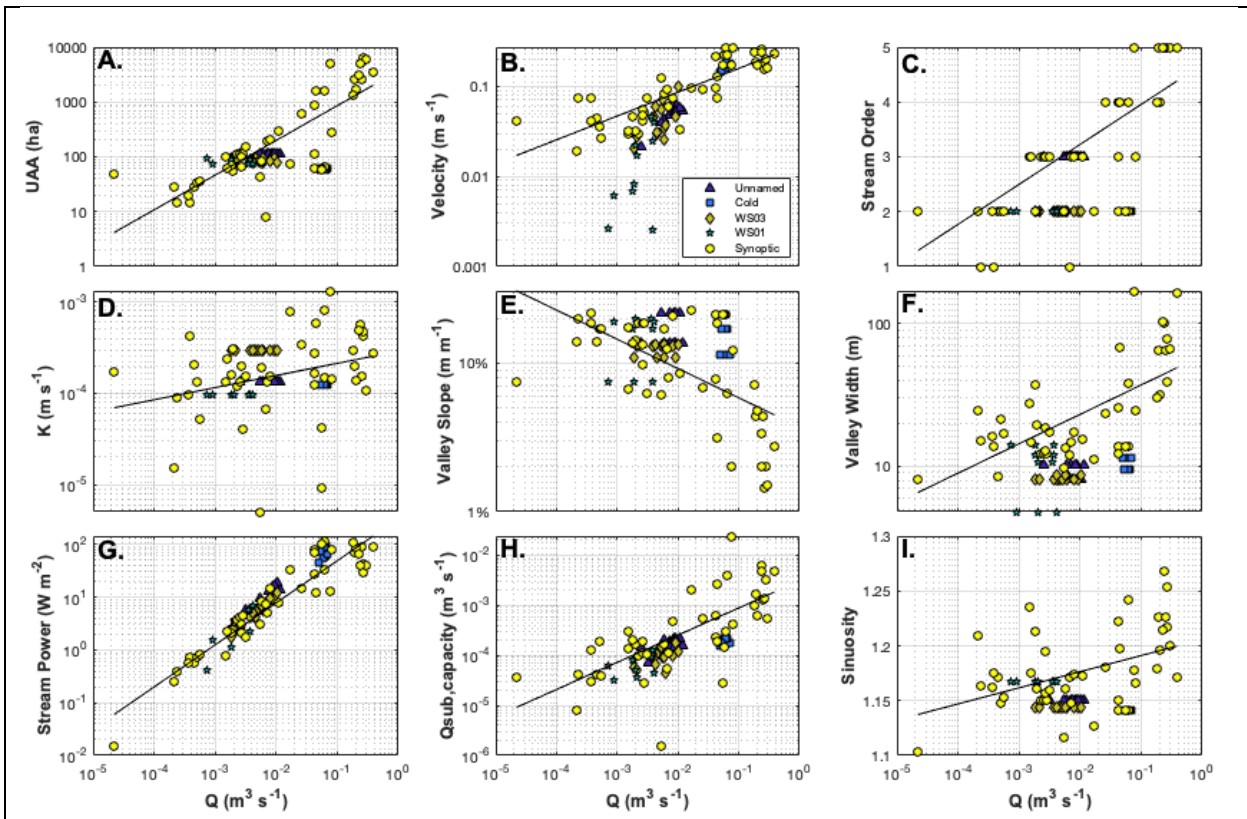

**Fig. 3.** For synoptic data (yellow circles), discharge exhibits a significant, monotonic trend with all other site variables considered (Mann-Kendall test; $p < 0.05$). Pairwise MK test results for all site characteristic pairs (i.e., all y-axis variables presented above) exhibit significant trends for all combinations ($p < 0.05$). The solid black line shows the best-fit power law regression for each panel. Data from unnamed creek (triangles, Cold creek (squares), WS03 (diamonds), and WS01 (stars) show the repeated injections through baseflow recession each headwater catchment. See supplemental Fig. S1 and S2 for similar plots with $HYP_{POT}$ and $UAA$ on x-axis.

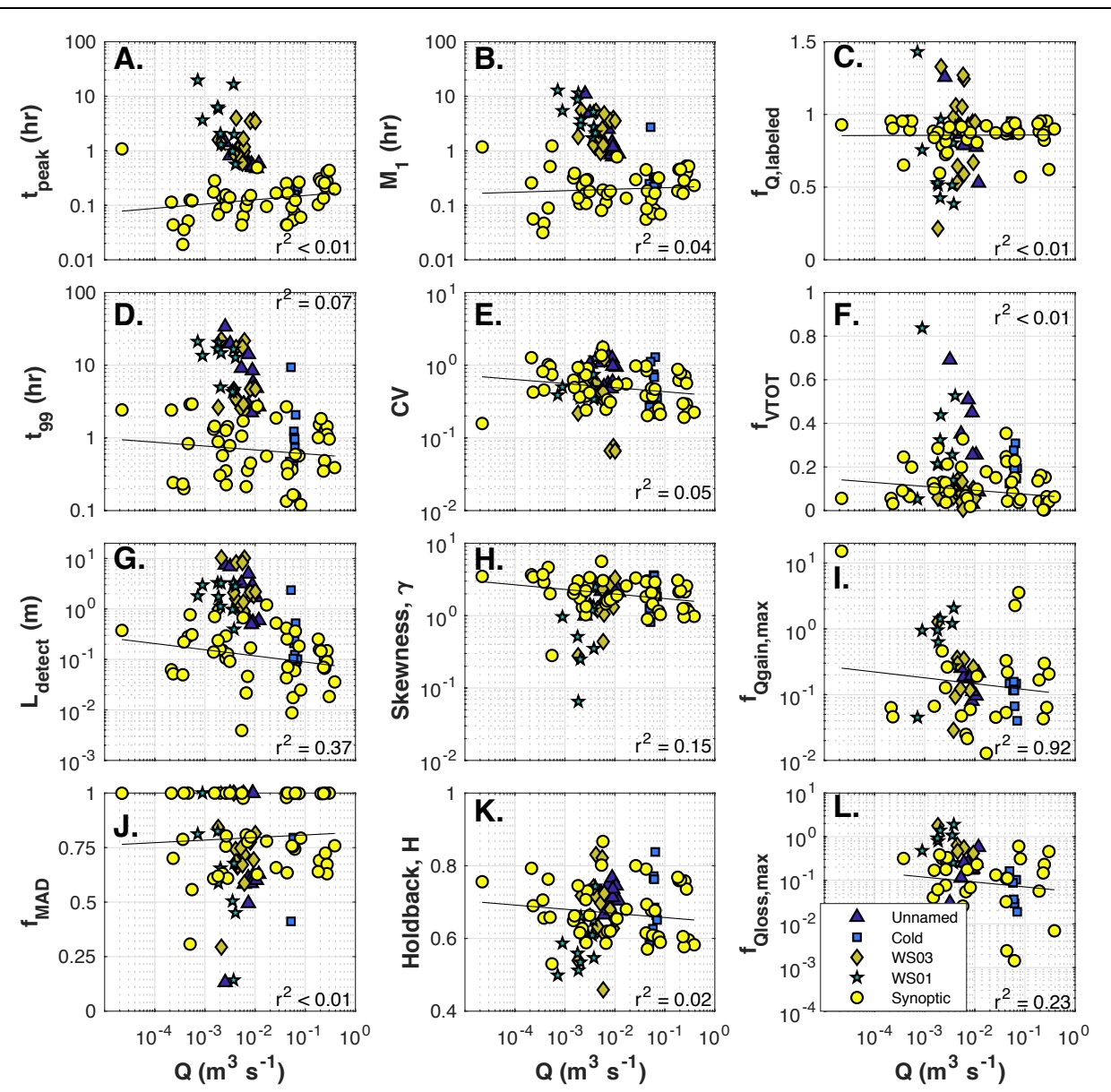

**Fig. 4. Fixed reach and synoptic data as a function of stream discharge. Statistical likelihood of significant relationships (Mann-Kendall test) and their direction (Sen's slope) are detailed for all sub-reaches and the synoptic data in Table 3. All trends shown here are significant (MK test, $p < 0.05$). The coefficient of determination for power law best-fits to synoptic data (black lines) are reported in Table 3. Data from unnamed creek (triangles, Cold creek (squares), WS03 (diamonds), and WS01 (stars) show the repeated injections through baseflow recession each headwater catchment. See supplemental Fig. S5 and S6 for similar plots with $HYP_{POT}$ and $UAA$ on x-axis.**

### 3.2 River corridor exchange trends with site characteristics

### 3.2.1 Basin-scale trends from synoptic campaign

An important element in our synoptic study design was the dynamic reach length, intended to minimize bias associated with the well-documented relationship between advective timescale and transient storage (e.g., Ward et al., 2013b; Schmadel et al., 2016a). Despite our efforts to hold advective travel time constant, we still found a trend of increasing $t_{peak}$ with increasing discharge in our synoptic study (Fig. 4A). Clearly, scaling reach length relative to the wetted channel width (20 wetted channel widths) is not a perfect solution.  A perfect experimental design would have resulted in no trend in advective time and provided a window of detection of constant size. While a trend was present, we also note that travel time based on $t_{peak}$ exhibits less variation than discharge (coefficient of variation 1.00 for travel time compared to 1.49 for discharge). For context, a recent study by Ward et al. (2018b) attempted to control for experiments with 20-min of advective time and accepted a range from 17 to 50 minutes as comparable. Thus, while our selection of study reach lengths was imperfect to achieve identical advective timescales, we contend that we have adequately controlled for advective time.

Overall we found significant trends (MK test; $p < 0.05$) between nearly all site characteristics and metrics describing river corridor exchange. Of the 130 pairings investigated, only three (stream order vs. $L_{detect}$; stream order vs. $f_{MAD}$; Sinuosity vs. $f_{Qlabeled}$) were not significant (Table 3). However, while network-scale trends do exist, we note high site-to-site variation in the data set as evidenced by the low $r^2$ for the power-law fits (see trendlines in Fig. 4), representative of the range of explanatory power observed. Across all 130 pairings investigated, we found very little explanatory value in the model fits, with a median $r^2$ of less than 0.03 (i.e., the variance in the model errors are about 3% less than the variance in the dependent variable itself). The lack of explanatory power for individual variables may indicate that fits based on more complex functional forms and/or multivariate approaches would increase predictive power. We did observe improved $r^2$ for all fits using both $Q$ and $HYP_{POT}$ compared to univariate regressions (Table S3).

**Table 3. Mann-Kendall tests indicate significant (p < 0.05), monotonic trends relating almost all site characteristics and metrics of river corridor exchange for the synoptic survey locations. The direction of the trend is indicated as increasing ("+") or decreasing ("(-)"). Three relationships that lacked a significant trend are denoted "?" in the table below. Additionally, the magnitude of the coefficient of determination ($r^2$) for univariate power-law fit is presented as an indicator of the power of a simple regression.**

| | Experimental Design | | | AD vs. TS | Short term storage | | | | rSAS | | | Long-term storage | | Metric |
|---|---|---|---|---|---|---|---|---|---|---|---|---|---|---|
| | $t_{99}$ | $t_{peak}$ | $L_{detect}$ | $f_{MAD}$ | $M_1$ | CV | $\gamma$ | Holdback | $f_{Q,labeled}$ | $f_{VSTR}$ | $f_{VTOT}$ | $f_{QGAIN,MAX}$ | $f_{QLOSS,MAX}$ | |
| Q | (-) | + | + | + | + | (-) | (-) | (-) | + | (-) | (-) | (-) | (-) | |
| UAA | + | + | + | (-) | + | (-) | (-) | (-) | (-) | (-) | (-) | + | (-) | |
| V | (-) | + | + | + | (-) | (-) | (-) | (-) | + | (-) | (-) | + | + | |
| Stream Order | + | + | ? | ? | + | (-) | (-) | (-) | (-) | (-) | (-) | + | (-) | |
| K | (-) | + | (-) | + | + | (-) | (-) | (-) | (-) | (-) | (-) | + | (-) | Sen's Slope |
| Down-Valley Slope | (-) | (-) | (-) | (-) | (-) | + | + | + | (-) | + | + | (-) | (-) | |
| Valley Width | + | + | + | + | + | (-) | (-) | (-) | + | (-) | (-) | (-) | (-) | |
| Stream Power | (-) | + | + | + | + | (-) | (-) | (-) | + | (-) | (-) | (-) | (-) | |
| $Q_{sub,cap}$ | (-) | + | (-) | + | + | (-) | (-) | (-) | (-) | (-) | (-) | + | (-) | |
| Sinuosity | + | + | (-) | + | + | + | (-) | + | ? | (-) | (-) | + | (-) | |
| Q | 0.07 | 0.00 | 0.37 | 0.00 | 0.04 | 0.05 | 0.15 | 0.02 | 0.00 | 0.15 | 0.00 | 0.92 | 0.23 | |
| UAA | 0.00 | 0.11 | 0.51 | 0.01 | 0.03 | 0.06 | 0.13 | 0.01 | 0.00 | 0.18 | 0.05 | 0.00 | 0.02 | |
| V | 0.18 | 0.00 | 0.29 | 0.02 | 0.08 | 0.08 | 0.03 | 0.01 | 0.02 | 0.19 | 0.01 | 0.01 | 0.00 | |
| Stream Order | 0.01 | 0.10 | 0.03 | 0.00 | 0.04 | 0.02 | 0.10 | 0.00 | 0.00 | 0.14 | 0.04 | 0.00 | 0.03 | |
| K | 0.02 | 0.03 | 0.10 | 0.02 | 0.00 | 0.13 | 0.10 | 0.04 | 0.01 | 0.07 | 0.01 | 0.03 | 0.11 | |
| Down-Valley Slope | 0.01 | 0.19 | 0.03 | 0.04 | 0.07 | 0.05 | 0.04 | 0.01 | 0.00 | 0.10 | 0.13 | 0.01 | 0.02 | |
| Valley Width | 0.00 | 0.02 | 0.01 | 0.01 | 0.01 | 0.06 | 0.16 | 0.03 | 0.00 | 0.06 | 0.09 | 0.92 | 0.34 | $r^2$ for power law fit |
| Stream Power | 0.07 | 0.00 | 0.74 | 0.00 | 0.04 | 0.05 | 0.15 | 0.02 | 0.00 | 0.15 | 0.00 | 0.92 | 0.23 | |
| $Q_{sub,cap}$ | 0.02 | 0.03 | 0.62 | 0.00 | 0.00 | 0.19 | 0.30 | 0.07 | 0.00 | 0.17 | 0.03 | 0.02 | 0.11 | |
| Sinuosity | 0.01 | 0.00 | 0.01 | 0.01 | 0.00 | 0.00 | 0.03 | 0.00 | 0.00 | 0.03 | 0.01 | 0.92 | 0.22 | |
| MAX | 0.18 | 0.19 | 0.74 | 0.04 | 0.08 | 0.19 | 0.30 | 0.07 | 0.02 | 0.19 | 0.13 | 0.92 | 0.34 | |
| MEDIAN | 0.02 | 0.03 | 0.20 | 0.01 | 0.04 | 0.05 | 0.11 | 0.02 | 0.00 | 0.14 | 0.02 | 0.02 | 0.11 | |
| MEAN | 0.04 | 0.05 | 0.27 | 0.01 | 0.03 | 0.07 | 0.12 | 0.02 | <0.01 | 0.12 | 0.04 | 0.37 | 0.13 | |
| MIN | <0.01 | <0.01 | 0.01 | <0.01 | <0.01 | <0.01 | 0.03 | <0.01 | <0.01 | 0.03 | <0.01 | <0.01 | <0.01 | |

### 3.2.2 Fixed reach vs. synoptic results

We found decreasing $t_{99}$ with increasing discharge for the synoptic study (Fig. 4D), which in turn resulted in a systematic reduction in the possible length of flowpaths that could be detected by tracer (Fig. 4G). Note that this ranges, on average, from 0.35 m at the lowest discharge to only 0.09 m at the highest discharge and the reach with the largest $L_{detect}$ was only 2.0 m. In contrast, reach lengths used in the fixed reach studies were much longer relative to stream size than the synoptic reaches, thus $t_{peak}$, $M_1$, $t_{99}$, and $L_{detect}$ were all much larger in the fixed reach studies (Table 4). These metrics all exhibited significant trends with discharge (Table 3), but the trends were not regularly consistent in their direction with the synoptic results. Overall, we found predominantly decreasing $t_{peak}$ with discharge in the fixed reaches - opposite to the synoptic finding - for 9 of 11 fixed reaches (and steeper Sen's slope in 9 of 11 fixed reaches). We also found decreasing $t_{99}$ with discharge in 9 of 11 fixed reaches (all with steeper Sen's slope than the synoptic), and decreasing $L_{detect}$ with discharge in 9 of 11 fixed reaches (all with steeper Sen's slope than the synoptic). Even with the longer reach lengths, relative to stream size, used

in the fixed reach studies, $L_{detect}$ averaged only ~2.0 m, and ranged from a maximum of 10 m to a minimum of 0.10 m.

With respect to short-term storage, we found increasing $M_1$ with increasing discharge in the synoptic study, but this direction was reflected in only 2 of 11 fixed reaches. Sen's slope was larger in magnitude for 10 of the 11 fixed reaches, indicating $M_1$ interpreted from the fixed reach approach is more sensitive to discharge than the synoptic approach. We found overall decreasing $CV$, $\gamma$, and $H$ with increasing discharge in the synoptic study, indicating a decreasing importance of non-advective processes in the downstream direction along the network. The direction of this trend is consistent with 7 fixed reaches for $CV$, 2 sites for $\gamma$, and 3 sites for $H$. Regardless of the direction of the relationship, the magnitude of Sens slope was larger for all fixed reaches compared to the synoptic study, indicating increased sensitivity to discharge relative to the synoptic sites.

For long-term storage and mass involved in advection-dispersion, we again found fixed-reach trends were steeper and often opposed the direction of the trend for the synoptic data. For the synoptic study we found decreasing $f_{Qgainmax}$ (Fig. 4I) and $f_{Qlossmax}$ (Fig. 4L) with increasing discharge, which is consistent with 5 and 6 of the 11 fixed reaches, respectively. For the synoptic study we found an overall decreasing $f_{MAD}$ with increasing discharge, consistent with 7 of the 11 fixed reaches. The magnitude of Sens slope was larger for the fixed reaches than the synoptic study for $f_{MAD}$, $f_{Qgainmax}$, and $f_{Qlossmax.}$

The SAS analysis revealed decreasing sampling of the total storage zone ($f_{Vtot}$) with increasing discharge, but increasing $f_{Q,labeled}$ with discharge for the synoptic study. Together, these results indicate that increasing discharge in synoptic experiments resulted in sampling a larger fraction of the water exiting the reach, but smaller total volume of storage. Put another way, experiments in locations with higher discharge were more likely to measure storage in (or proximal to) the stream channel at the expense of measuring more distal flowpaths and less-connected storage. For the fixed reach studies, we found decreasing $f_{Vtot}$ and $f_{Qlabeled}$ in 7 and 6 of the 11 reaches, respectively. In all cases, the magnitude of Sens slope was larger for the fixed reaches than the synoptic study.

Table 4. Sen's slope for all discharge-metric relationships across fixed reach study sites and the synoptic site data. All relationships were significant (p < 0.05) using the Mann-Kendall test. The values shown indicate the direction of the relationship based on a Sen's slope estimator ("+" indicates a direct relationship with discharge, and "(-)" indicates inverse relationship with discharge). Slopes were larger in magnitude for the fixed reaches in all cases except Cold Creek sites 12 and 23 for $t_{peak}$ and Cold Creek site 12 for $M_1$, denoted with "*".

| Catchment | Study Reach | n | Experimental Design | | | AD vs. TS | Short term storage | | | | rSAS | | | Long-term storage | |
|---|---|---|---|---|---|---|---|---|---|---|---|---|---|---|---|
| | | | $t_{99}$ | $t_{peak}$ | $L_{detect}$ | $f_{MAD}$ | $M_1$ | $CV$ | $\gamma$ | Holdback | $f_{Q,labeled}$ | $f_{VSTR}$ | $f_{VTOT}$ | $f_{QGAIN,MAX}$ | $f_{QLOSS,MAX}$ |
| Unnamed | Site12 | 6 | (-) | (-) | + | + | (-) | (-) | + | + | (-) | (-) | (-) | + | (-) |
| | Site23 | 6 | (-) | (-) | + | (-) | (-) | (-) | + | + | (-) | (-) | (-) | + | (-) |
| Cold | Site12 | 5 | (-) | (-)* | + | (-) | (-)* | (-) | (-) | (-) | + | + | + | (-) | + |
| | Site23 | 5 | + | (-)* | (-) | + | + | + | + | + | + | (-) | (-) | + | + |
| WS03 | Site12 | 5 | + | (-) | (-) | (-) | + | (-) | + | + | + | (-) | + | (-) | + |
| | Site23 | 5 | (-) | (-) | + | + | (-) | (-) | + | (-) | (-) | (-) | + | + | (-) |
| | Site34 | 5 | (-) | + | + | + | (-) | (-) | (-) | (-) | (-) | (-) | (-) | (-) | (-) |
| WS01 | Site12 | 3 | (-) | (-) | + | (-) | (-) | (-) | + | + | (-) | + | + | + | (-) |
| | Site23 | 2 | (-) | + | + | (-) | (-) | + | + | + | + | + | (-) | (-) | + |
| | Site34 | 2 | (-) | (-) | + | (-) | (-) | + | + | + | + | (-) | (-) | (-) | + |
| | Site45 | 3 | (-) | (-) | + | (-) | (-) | + | + | + | + | + | (-) | (-) | + |
| HJA | Synoptic | 46 | (-) | + | + | (-) | + | (-) | (-) | (-) | + | (-) | (-) | (-) | (-) |

## 3.3 Selection of study reach length across the network

For the injections that specifically tested the study reach length, we found the most consistent advective timescales were obtained by scaling reach length to 20 times wetted channel width (Fig. 5). Ranges of advective timescales were 25.2 minutes for the fixed-length approach, 27.2 minutes for the fixed timescale approach, and 4.8 minutes for the 20× wetted channel width approach (Fig. 5A). It is notable that our estimates of a 10-min advective time were reasonably accurate for the three highest-discharge reaches, but the lowest-discharge replicate primarily drives the visually steep trend. We hypothesize that a better estimate of advective velocity -- such as using a dye tracer rather than following debris or a longer length-scale of integration -- may have improved that estimate. For $t_{99}$, ranges for the 10-min and 150-m approaches are about 29% and 22% larger, respectively, than the 20× wetted channel width approach (Fig. 5B). Differences are even more striking for other parameters, with the 10-min and 150-m study designs, yielding 147% and 93% larger ranges for $H$ compared to the 20× wetted channel width approach (Fig. 5C). Similarly, the 10-min and 150-m approaches result in ranges of $\gamma$ that are 96% and 101% larger than the ranges using the 20× wetted channel width approach (Fig. 5D).

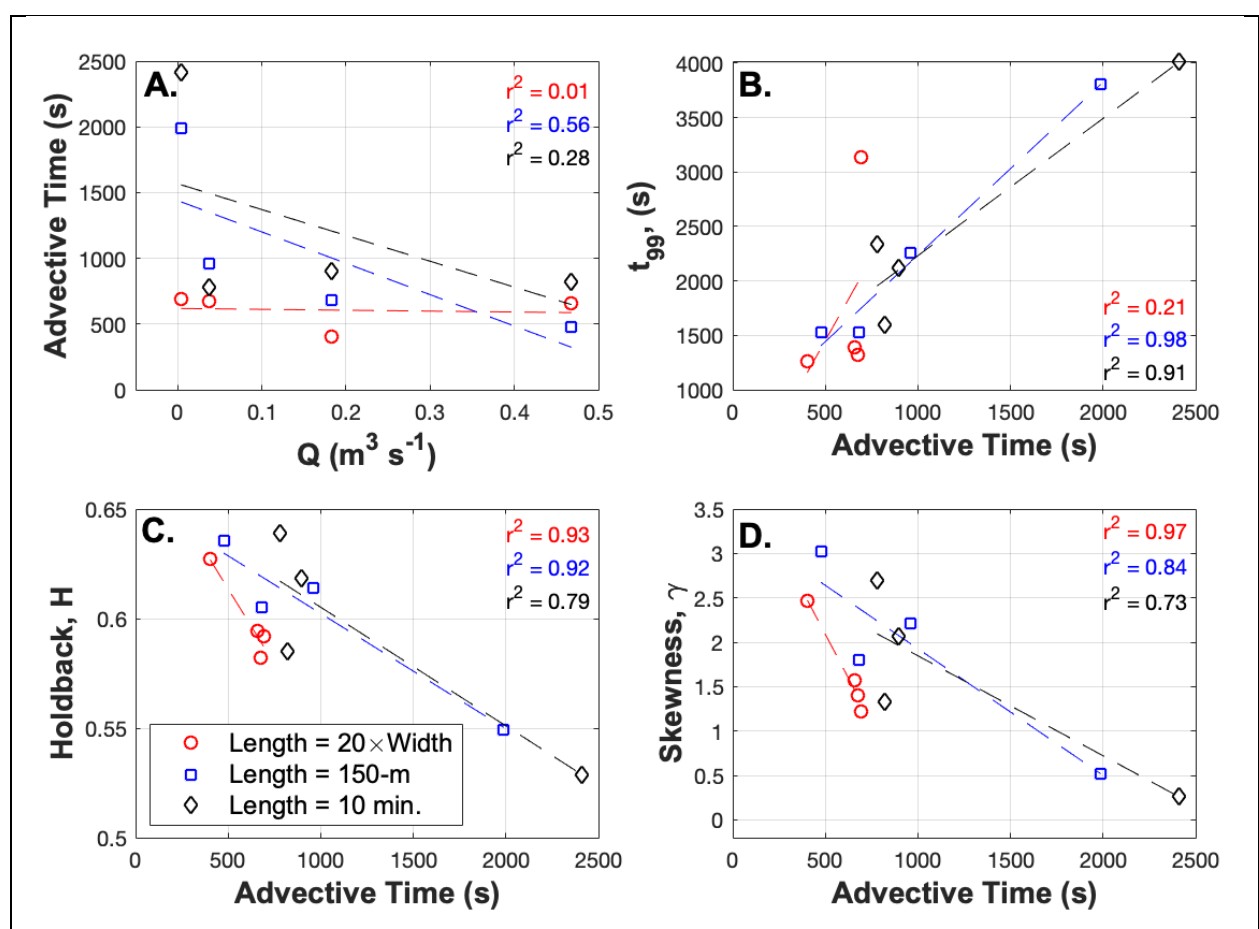

Fig. 5. Comparison of fixed reach (150-m), adaptive reach length (20 wetted channel widths), and fixed advective time (10-min) approaches for standardization of stream solute tracer studies. (A) Control of advective time across four stream orders. Additional panels show the observations and a best-fit linear regression for (B) longest detection timescale, (C) holdback, and (D) skewness in relation to the advective time of the study. Best-fit linear regressions are shown as dashed lines in each panel.

## 4 Discussion

### 4.1 How do discharge and local geomorphic setting modulate river corridor exchange?

Our overarching objective in this study was to test the conceptual model of Wondzell (2011), which predicted systematic changes in river corridor exchange as a function of changing baseflow and geomorphic setting (Fig. 6A). We found a generally decreasing influence of river corridor exchange with increasing steady-state discharge through space for most metrics considered (Fig. 4; Fig. 6B-F). While we could not measure exchange flux directly, we find $t_{99}$, $f_{MTS}$, $f_{VTOT}$, and $f_{QLOSS,Max}$ generally decrease in parameter value with increase in catchment area (Figs. 6B-F). This finding is in agreement

with the conceptual model of Wondzell (2011), who predicted $Q_{HEF}/Q$ would decrease as drainage area increased. We did find an increasing fraction of total discharge sampled in higher discharge locations (Fig. 4C), but the overall trend indicates that $Q_{HEF}$ does not grow as rapidly as $Q$, moving downstream along the network. This is consistent with findings of decreased river corridor exchange in network
locations with larger discharge (e.g., Covino et al., 2011; Ward et al., 2013c).

Two explanations have been posed relating river corridor exchange to time-variable baseflow in a given study reach, both of which result in less exchange under higher discharge conditions. First, many conceptual models would predict that increasing baseflow is associated with increasing groundwater
discharge to the stream, resulting in compression of hyporheic zones and decreased river corridor exchange (Hakenkamp et al., 1993; Hynes, 1983; Palmer, 1993; Vervier et al., 1992; White 1993). Second, exchange may change little during storm events because, under a wide range of discharge conditions, the effect of the geomorphic features driving exchange flows may be relatively static (Ward et al., 2017b). Thus, if $Q_{HEF}$ is relatively static, as $Q$ increases the relative amount of relative exchange
($Q_{HEF}/Q$) will decrease. Both explanations appear logical and suggest that river corridor exchange should change systematically with discharge. However, we did not find a consistent pattern in our synoptic field study. Rather, of the diverse array of metrics used to characterize river corridor exchange in the synoptic study, some increased and some decreased with increasing discharge. We found similarly contradictory results among our fixed reach studies. For example, only 2 of 11 fixed reaches
exhibited the expected negative relationship based on skewness (one indicator of $Q_{HEF}/Q$) and discharge (Table 4).

**4.2 Heterogeneity in the river network**

Wondzell's (2011) conceptual model followed general predictions about systematic changes in channel
morphology with increasing stream size, predicting channel width, channel depth, and flow velocity will all increase with discharge, both over time at a fixed cross section or with location at a given time within a stream network. Further, bed sediment size distributions would generally decrease in a downstream direction (see, for example, Leopold and Maddock, 1953). While the physical attributes we measured at our synoptic sites did show systematic variation, the pattern in saturated hydraulic
conductivity ($K$) was contrary to expectations, as we found $K$ increased in the downstream direction (Fig. 3D). This change was so large that it overwhelmed the effect of decreasing longitudinal gradient so that the hyporheic potential actually increased in a downstream direction. We note, however, that our studies only spanned about 4 orders of magnitude in hyporheic potential while Wondzell's (2011) model visualizes a range that spans 14 orders of magnitude. Our study is also limited to the upper end of
the range in hyporheic potential depicted by Wondzell (2011).

Our dataset also showed substantial spatial heterogeneity in all metrics along the river corridor. While Wondzell's (2011) conceptual model does not expressly disallow such heterogeneity, the data points he used to develop the conceptual model suggest very uniform changes with watershed area and little
change in hyporheic potential from 2nd- to 5th-order reaches within the same mountain stream network

studied here. Our results suggest that the influence of reach scale heterogeneity among sites may be as large as, or even larger than, the expected systematic changes with watershed size. We also note that our results may differ from those of Wondzell (2011) for methodological differences. First, Wondzell (2011) based his estimates of $K$ from extensive well networks at each of his sites, using the geometric mean of all wells – including many wells on the floodplain adjacent to the stream as well as piezometers installed through the streambed. This study estimated $K$ from a single, 50-cm deep piezometer located in the channel thalweg and Wondzell's (2011) data show that $K$ is higher in piezometers inserted into the shallow streambed than in floodplain sediment adjacent to the stream. Second, Wondzell (2011) used numerical simulations from groundwater flow models to calculate $Q_{HEF}$, whereas exchange metrics in this study were derived from stream solute tracer injections. Solute injections are sensitive to both surface (in-stream) and subsurface transient storage, and metrics derived from these studies have a known bias toward the shortest transit times (Harvey et al., 1996; Wagner and Harvey, 1997; Harvey and Wagner, 2000), a bias that is clearly evident in our data. For example, the longest timescale flowpath detectable, interpreted from $t_{99}$, in our study reaches ranged from about 8 minutes to 2.8 hours. In contrast, Wondzell's (2011) simulations included flowpaths with up to 10 day transit times. However, cell sizes in the finite-difference grids used in his models limited the shortest flow paths that could be simulated, so his estimates of $Q_{HEF}$ should underrepresent the very shortest flow paths present within the reach.

Transient storage in the surface (in-stream) channel is known to influence tracer breakthrough in solute injection experiments and more specifically, has been documented in our study basin (*Jackson et al.*, 2012, 2013). Thus, our data represent a combination of surface and hyporheic transient storage, but we expect the hyporheic component will be most sensitive to hydraulic conductivity. Thus, deviation from the expected trend with hyporheic potential may simply indicate that our tracer studies were not solely representative of $Q_{HEF}$ between a stream and its hyporheic zone as defined and assumed by Wondzell (2011) (Fig. 6). Our SAS analyses indicate we measured storage volumes larger than the stream in most reaches, but it is unclear what the mechanisms or timescales of exchange were for the storage locations measured. Overall, this unique basin scale dataset does not appear to support Wondzell's (2011) conceptual model with respect to hyporheic potential, but it does not disprove it either due to the limitations in methods and clustering on only the highest end of the axis likely biased our results. Still, we suggest local-scale processes specific to individual sites may overwhelm basin-scale trends and limit the ability of continuum based conceptual models, such as Wondzell (2011), to predict local-scale hyporheic and river corridor exchange dynamics.

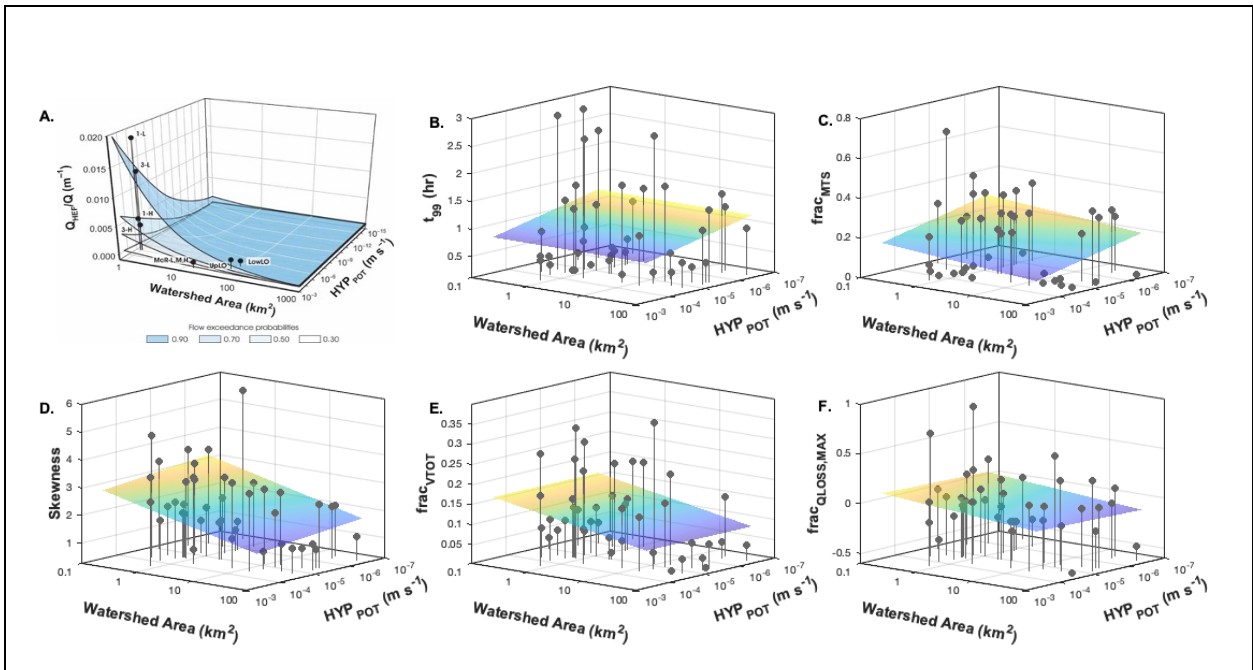

**Fig. 6. Comparison of (a) conceptual model of river corridor exchange (reprinted from _Wondzell_ (2011) with permission) and findings from this study including a best-fit planar surface fit to the synoptic data for each panel (dots show the data points, stems extend to the bottom X-Y plane to aid in visualization; planar surface light-to-dark shading indicate high-to-low for the Z-axis variable). Panels show trends for a sub-set of variables representing (B) experimental design, (C) separation of advection-dispersion from transient storage, (D) short-term storage, (E) StorAge Selection, and (F) long-term storage. Goodness of fit and slopes for each fit are summarized in Table S3.**

**4.3 Can space-for-time or time-for-space relationships be used to transfer findings based on reach-scale characteristics?**

Transferability of findings in space or time relies upon two assumptions, both of which are necessary conditions for reliable prediction. First, transferability requires that the process of interest varies
5  systematically with at least one observable variable at the study and predicted sites. In our case, this requires the relationship between discharge and river corridor exchange to be measurable and robust, commonly judged on the basis of a goodness-of-fit metric for a regression. Transferability also requires that the functional form established from the observations holds for the conditions that are being predicted. In the temporal domain this is most commonly interpolation in time to predict river corridor
10  exchange under a discharge condition that was not actually observed (e.g., Harman et al., 2016; Ward et al., 2018a). In the spatial domain, this transferability strategy may manifest as interpolation between observed sites (e.g., Covino et al., 2011; Mallard et al., 2014) or extrapolation to sites that are morphologically similar, such as extending findings from one headwater site to make predictions in an adjacent basin or another stream reach (e.g., Jencso et al., 2011; Covino et al., 2011; Stewart et al.,
15  2011). This approach assumes that the relationship holds because the observational and predicted sites

are similar. However, we find that there is substantial variation among sites, particularly when reaches of similar size yield opposing relationships with explanatory variables (Tables 2, 3).

Overall, we conclude that discharge alone is a poor predictor of river corridor exchange in mountain stream networks due to heterogeneity in reach-scale geomorphic setting and should not be used as the sole basis for spatial or temporal extrapolation of findings. We found opposing relationships between river corridor exchange and discharge through space (synoptic approach) and time (fixed reach approach). For all metrics considered, at least 18% (2 of 11) of the intensively studied fixed reaches had trends opposite of that what would be predicted from the one-time sampling of the synoptic study. Moreover, the opposing trends were always located across at least two different landform types, and there were examples of within-landform type disagreement for every metric considered. Furthermore, the regressions we developed indicated that there was substantial inter-site heterogeneity overriding the observed network-scale trends. These findings are useful for identifying best practices to ultimately develop better scaling relationships to predict river corridor exchange as a function of hydrologic forcing and geomorphic setting from headwaters to oceans. For example, intensively studying a small number of study reaches is not indicative of the conditions occurring across an entire basin, even at the scale of our 5th order basin. We further develop suggestions for best practices and considerations in the next section.

**4.4 Best practices to measure and interpret exchange-discharge relationships**

Stream solute tracers are perhaps the empirical method most frequently used to measure river corridor exchange. Given the relative ease and low cost of this method, it is unsurprising that many studies have used solute tracer studies under different discharge conditions to assess relationships between discharge and river corridor exchange. For example, some studies repeat solute injections in a fixed reach under range of discharge conditions during different seasons (e.g., Zarnetske et al., 2007; Ward et al., 2018b), during baseflow recession (e.g., Payn et al., 2009; Ward et al., 2012), or during storm events (e.g., Ward et al., 2013b; Dudley-Southern and Binley, 2015). Still others use spatial replication at multiple sites within a network to construct a relationship that can be used to predict behavior for unstudied reaches during a single discharge condition (e.g., Jencso et al., 2011; Covino et al., 2011; Stewart et al., 2011). However, limitations of stream solute tracers are well documented in the literature as mentioned above (Harvey et al., 1996; Wagner and Harvey, 1997; Harvey and Wagner, 2000; Drummond et al., 2012; Kelleher et al., 2013; Ward et al., 2017a).

The ability to detect late-time tailing of the tracer (e.g., Drummond et *al.*, 2012) and parameter dependence on advective timescales of transport (e.g., Schmadel et al., 2016a) limit the interpretability of solute tracer studies. However, armed with a seemingly straightforward tool (e.g., stream solute tracers) and the expectation to find trends with discharge, it is logical that many studies have concluded discharge (or its tightly correlated proxy of drainage area) is a meaningful predictor of river corridor exchange. However, we argue this may be a self-fulfilling prophecy as it is often unclear exactly what is being measured by the tracer observations. For fixed-reach studies repeated under different discharge

conditions, the observed trends between river corridor exchange and discharge can be plausibly explained by either physical transport processes or simply limitations of the tracer method. Indeed, this unfortunate conclusion was clearly illustrated by recent studies focused on solute tracer studies across a range of discharge conditions (e.g., Wondzell, 2006; Schmadel et al., 2016a). Thus, we contend that it is unknown if reported trends in the literature reflect mechanistic understanding of the river corridor or suffer from confirmation bias. Therefore, we detail two best practices for conducting and interpreting stream solute tracer tests for those seeking to do as we have attempted in this study.

### 4.4.1 Best practice 1: Control for advective timescales instead of reach length.

The most common paradigm in stream solute tracer studies is to use a fixed-length study reach, and hold length constant to compare different reaches (e.g., Payn et al., 2009; Covino et al., 2011) or to compare different discharge conditions at a single reach of fixed length (e.g., Schmadel et al., 2016a; Ward et al., 2013a). The implicit logic is that by fixing the reach length, the same morphologic features interact with the tracer and allow the researcher to measure changes in the same processes. However, this is only true in the case where the same suite of flowpaths can be detected. When advective timescales decrease, the window of detection (i.e., the longest timescale flowpath that can be detected) should decrease in response (e.g., Schmadel et al., 2016a). As a result, the fixed reach causes systematic bias in the tracer experiment. Higher discharges will have smaller windows of detection, biasing the results toward shorter timescale flowpaths compared to low-discharge injections.

Based on our findings (Fig. 5), plus the well-documented interaction of advective timescale with river corridor exchange measured with solute tracers, we strongly recommend experimental designs that control for advective timescale. We suggest that an upstream location be established and fixed in space. Then the length of the study reach should be determined, either by scaling by channel width (e.g., 20 times the wetted channel width) or by using a dye tracer to measure advective velocity over a length equal to perhaps 10 wetted channel widths, and then using advective velocity to calculate a study reach length that provides uniform advective travel times in all reaches studied.

When tracer injections are designed to provide uniform advective travel times, the resulting study reach lengths will be longest in the largest streams and/or at times of high discharge; reaches will be shortest under low discharge conditions. It is critical that the shortest reach length still encompass a length of stream that is sufficient to integrate representative variation in morphology of the study system. If reaches are too short, high reach-to-reach variability will be generated by one or a few morphologic features and these local conditions are likely to dominate comparisons among reaches and make it difficult to discern the influence of changing hydrologic conditions. It will be difficult to determine a length-scale long enough to integrate the full range of morphologic features present in any given stream. Schmadel et al. (2014) suggested that a morphologically representative reach could be determined by knowing the length of spatial autocorrelation of morphologic features, but this requires substantial effort to survey or map the study reach prior to conducting a tracer test. A less effort-intensive, but more equipment-intensive approach would be to place multiple sensors in the study reach (perhaps 10, 20, 35,

50, 75, and 100 wetted channel widths) and select most appropriate downstream breakthrough curves to compare based on similarity of advective timescales after conducting the tracer test.

It is also essential that measures of the advective timescale and window of detection be reported for each tracer test. For slug injections these would include $t_{peak}$ and $t_{99}$. For constant rate injections these would be time to the steepest point on the rising limb, time to median arrival ($M_1$), and time to achieve plateau. The $L_{detect}$ estimates should also be reported and these should be based on time to achieve plateau as that indicates when tracer has traveled the full length of all measurable flowpaths and only tracer-labeled water is being returned to the stream. These metrics describing the advective timescale are necessary both to confirm that comparisons among reaches in any given study are valid and to facilitate comparisons of results among published studies.

We acknowledge here that the steps we've recommended above will require substantial time and analysis to design a stream tracer experiment. However, we contend this additional work is necessary to maximize the interpretability of the data and enable meaningful comparison across space and time.

### 4.4.2 Best Practice 2: Critical evaluation of which flowpaths may have been measured by the experiment

One persistent limitation of interpreting stream solute tracers is the inability to know which flowpaths and features were actually measured in the study reach. While additional observations in storage zones have been attempted via monitoring wells or geophysical imaging, multiple studies show that solute observed in the storage zone itself is not necessarily meaningful, as the stream breakthrough curve integrates only a sub-set of flowpaths (Ward et al., 2010a, 2017b, Toran et al., 2012, 2013). Briggs et al. (2009) suggest additional measurements in the surface storage domain may allow for parsing surface from subsurface transient storage. However, this approach relies upon measurement of a representative in-stream storage zone and interpretation via the transient storage model, which is known to be limited in identifiability of parameters and transferability to other sites (e.g., Kelleher et al., 2013; Ward et al., 2017a).

One simple approach to estimate the spatial and temporal scales of the measured flowpaths is to consider the truncation of the breakthrough curve itself. The window of detection describes the longest flowpath timescale that may have been measured. Several studies have converted this timescale to a length scale using Darcy's Law, parameterized it with representative values for hydraulic conductivity, porosity, and valley slope as a proxy for hydraulic gradient (after Ward et al., 2017b; 2018a). While imperfect, this interpretation at least indicates a spatial scale of flowpaths that may have been observed. For example, in previous studies of a small stream in the HJA basin (WS01; Fig. 1), where extensive penetration of the tracer into the subsurface was documented across a 10+ m wide valley bottom (Voltz et al., 2013; Ward et al., 2017b), the longest flowpaths detected by a tracer returning to the stream still only averaged 0.21 m (range 0.004 to 1.2 m) compared to overall reach lengths of tens of meters. This means that these studies were measuring in-stream storage and only the shortest and fastest subsurface flowpaths -- not integrating all the exchange in the valley bottom.

The SAS approach implemented in this study provides some valuable additional, contextual information about the storage volume and discharge that inform interpretation of findings. For example, our synoptic study labeled an average of 86% of the outflowing discharge in the surface channel (range 57% to 95%). Still, this equated to having only sampled an average of 12% of the total storage volume in the reach (range 0.3% to 35%), suggesting a bias toward in-stream storage. This bias is confirmed by the realization that, on average, only 18% of tracer mass was involved in transient storage (range 0% to 69%). Hence, the SAS approach gives us additional insights and reveals biases in the tracer methods. Altogether, this study clearly indicates that multiple data collection, analysis, and modeling techniques are needed to develop scaling relationships representative of river corridor exchange across varying hydrologic forcing and geomorphic settings.

## 5 Conclusions

We set out to leverage novel data sets collected across a 5[th] order basin to test the existence of systematic relationships linking river corridor exchange with temporal variation in discharge, spatial patterns in discharge, and local geomorphic setting. We specifically intended to use these data to critically test Wondzell's (2011) conceptual model (Fig. 6A). We found systematic patterns, namely decreases in several indicators of river corridor exchange with increasing discharge in space (i.e., moving downstream in the network), confirming this part of the Wondzell (2011) conceptual model. Wonzell's (2011) model predicts the same trend for increasing baseflow discharge in time, but we found both direct and inverse relationships between river corridor exchange and discharge at fixed reaches under varied baseflow conditions. These findings reflect a high degree of heterogeneity on a reach-to-reach basis in space, likely overwhelming or obscuring river corridor exchange patterns that might emerge in more spatially continuous and larger scale assessments, which would be a better test of the Wondzell (2011) model. Importantly, we document consistent trends with discharge that have low explanatory power (low $r^2$) despite being statistically significant in their direction, indicating that we have little predictive power. Moreover, our findings reveal the challenges that must be addressed to design and interpret stream solute data among sites or discharge conditions. Finally, we did not confirm Wondzell's (2011) predicted pattern with respect to local hyporheic potential at a site, which may have been confounded by integration of both surface and hyporheic storage by the stream solute tracers or by local-scale heterogeneity not captured in our reach-scale site characterization. Collectively, the larger Sen's slopes for the fixed reaches, when compared across variable hydrologic conditions, may indicate more temporal variation at a site through the season than there is through the network under a the single baseflow condition. This means that caution is needed in applying synoptic sampling approaches across time when studying river corridor exchange conditions in a river network.

This study documented the interaction between advective travel times and measurement of river corridor exchange with solute tracers. Our synoptic study design controlled for this complication by scaling study reach lengths based on wetted channel width. For future studies focused on exchange-

discharge relationships, we suggest two best practices. First, controlling for advective time to measure consistent timescales of storage processes and limit artifacts that are due to limitations of solute tracer studies. Second, we suggest analyses that focus on the fractions of storage volume and outflow that were labeled with tracer to provide context for interpreting recovered timeseries. We also note that many previous studies have relied upon small sample sizes and focused on singular explanatory variables of interest considered in isolation. We suggest this is primarily descriptive, and conclude that consideration of multiple, interacting controls will be necessary to achieve predictive understanding of river corridor exchange across varying hydrologic forcing and geomorphic setting from headwaters to large river networks.

Finally, we underscore that a one-time synoptic sampling campaign does not address local-scale variability that is created by variable discharge conditions, nor does extensive study of a single reach provide data that are reflective of variation in space in the river network. In short, space-for-time and time-for-space substitutions based on the methods used in our study are not a reliable basis for transferability nor prediction.

## 6 Summary of symbols used in this manuscript

| Symbol | Definition |
| --- | --- |
| $Q$ | Stream discharge ($m^3$ $s^{-1}$) |
| $Q_{HEF}$ | Hyporheic exchange flux ($m^3$ $s^{-1}$) |
| $UAA$ | Upslope accumulated area (ha) |
| $S_{val}$ | Valley slope (m $m^{-1}$) |
| $Sinuosity$ | Sinuosity, stream length per planform length (m $m^{-1}$) |
| $K$ | Hydraulic conductivity (m/s) |
| $r$ | Radius of well casing (m) |
| $R$ | Radius of well screen (m) |
| $L_e$ | Screened length of well (m) |
| $T_0$ | Time for head to fall to 37% of initial value (s) |
| $Q_{sub,cap}$ | Down-valley capacity of the subsurface ($m^3$ $s^{-1}$) |
| $b_{valley}$ | Valley width (m) |
| $h_{valley}$ | Valley sediment depth (m) |
| $HYP_{POT}$ | Hyporheic potential (m/s) |
| $\Omega$ | Stream power (W $m^{-2}$) |
| $\rho$ | density of water (kg $m^{-3}$) |
| $g$ | Gravitational acceleration (9.81 m $s^{-2}$) |
| $S$ | Channel slope (m $m^{-1}$) |
| $v$ | Advective velocity (m $s^{-1}$) |
| $A$ | Cross-sectional area of stream ($m^2$) |
| $Q_{US}$ | Discharge at upstream end of study segment via dilution gauging ($m^3$ $s^{-1}$) |

| | |
|---|---|
| $Q_{DS}$ | Discharge at downstream end of study segment via dilution gauging (m$^3$ s$^{-1}$) |
| $C_{obs}(t)$ | Observed, background-corrected in-stream solute tracer concentration (g m$^{-3}$) |
| $t_{peak}$ | Time for concentration peak to advect through study reach (s) |
| $t_{99}$ | Time at which 99% of total recovered solute tracer mass occurs (s) |
| $t$ | Time elapsed since injection (s) |
| $C_{ADE}(t)$ | Concentration timeseries at downstream end of study reach estimated using advection-dispersion equation (g m$^{-3}$) |
| $M_{REC}$ | Tracer mass recovered from upstream injection at downstream monitoring location (g) |
| $D$ | Longitudinal dispersion coefficient (m$^2$ s$^{-1}$) |
| $C_{AD}(t)$ | Concentration time-series for a solute that is predominantly transported by advection and dispersion (g m$^{-3}$) |
| $M_{AD}$ | Mass primarily associated with transport by advection and dispersion (g) |
| $C_{TS}(t)$ | Concentration time-series for a solute that is predominantly associated with transient storage (g m$^{-3}$) |
| $M_{TS}$ | Mass primarily associated with transient storage (g) |
| $f_{MAD}$ | Fraction of mass primarily associated with advection and dispersion |
| $f_{TS}$ | Fraction of mass primarily associated with transient storage |
| $c(t)$ | Normalized, background-corrected concentration timeseries |
| $M_1$ | First temporal moment (s) |
| $\mu_n$ | n$^{th}$ order central temporal moment (s$^n$) |
| $CV$ | Coefficient of variation |
| $\gamma$ | Skewness |
| $H$ | Holdback |
| $F$ | Variable of integration used in calculation of holdback |
| $\tau$ | Time variable for integration used in calculation of holdback |
| $L_{detect}$ | Longest flowpath detectable based on a Darcy's Law estimate of hyporheic pore water velocities |
| $\theta$ | Porosity |
| $T$ | Age of water (s) |
| $p_Q(T)$ | Probability density of the forward transit time distribution |
| $M_{US}$ | Mass of solute tracer injected at the upstream end of the study reach (g) |
| $P_Q(T)$ | Cumulative form of the probability density of the forward transit time distribution |
| $Q_T(T)$ | Age-ranked discharge (m$^3$ s$^{-1}$) |
| $S_T(T)$ | Age-ranked storage (m$^3$) |
| $Q_{comp}(T)$ | Complement to the age-ranked discharge |
| $S_{comp}(T)$ | Complement to the age-ranked storage |
| $S_{ref}$ | Estimated total volume of storage in the study reach |
| $f_{VTOT}(T)$ | Fraction of total storage volume labeled by tracer |
| $f_{Q,labeled}(T)$ | Fraction of outflowing discharge labeled by tracer |

| | |
|---|---|
| $f_{VSTR}(T)$ | Fraction of stream volume labeled by tracer |
| $M_{LOSS}$ | Mass loss between upstream injection and downstream observation location (g) |
| $Q_{LOSS,MAX}$ | Maximum gross outflow of discharge along the study reach (m$^3$ s$^{-1}$) |
| $Q_{GAIN,MAX}$ | Maximum gross outflow of discharge along the study reach (m$^3$ s$^{-1}$) |
| $\Delta Q$ | Net change in discharge along the study reach (m$^3$ s$^{-1}$) |
| $f_{QGAIN,MAX}$ | $Q_{GAIN,MAX}$ as a fraction of $Q_{US}$ |
| $f_{QLOSS,MAX}$ | $Q_{LOSS,MAX}$ as a fraction of $Q_{US}$ |

**Acknowledgements**

Funding for this research was provided by the Leverhulme Trust (*Where rivers, groundwater and disciplines meet: a hyporheic research network*), the UK Natural Environment Research Council (Large woody debris – A river restoration panacea for streambed nitrate attenuation? NERC NE/L003872/1), and the European Commission supported *HiFreq: Smart high-frequency environmental sensor networks for quantifying nonlinear hydrological process dynamics across spatial scales* (project ID 734317), the US Department of Energy (DOE) Office of Biological and Environmental Research (BER) as part of Subsurface Biogeochemical Research Program's Scientific Focus Area (SFA) at the Pacific Northwest National Laboratory (PNNL). Data and facilities were provided by the HJ Andrews Experimental Forest and Long Term Ecological Research program, administered cooperatively by the USDA Forest Service Pacific Northwest Research Station, Oregon State University, and the Willamette National Forest and funded, in part, by the National Science Foundation under Grant No. DEB-1440409. Ward's time in preparation of this manuscript was supported by the University of Birmingham's Institute of Advanced Studies. Additional support to individual authors is acknowledged from National Science Foundation (NSF) awards EAR 1652293, EAR 1417603, and EAR 1446328. Ward and Wondzell were also supported by Department of Energy award DE-SC0019377. Any use of trade, firm, or product names is for descriptive purposes only and does not imply endorsement by the US Government. Any opinions, findings, and conclusions or recommendations expressed in this material are those of the authors. Finally, the authors acknowledge this would not have been possible without support from their home institutions. All data used in this study are archived in the Consortium of Universities for the Advancement of Hydrologic Science, Inc. (CUAHSI) HydroShare data repository, publicly accessible as https://doi.org/10.4211/hs.f4484e0703f743c696c2e1f209abb842

**Author Contributions**

Author contributions are detailed below using the Contributor Role Taxonomy (CRediT) scheme. Conceptualization, ASW, SMW, JPZ, NMS; methodology, all co-authors; formal analysis, ASW; investigation, ASW, NMS, JPZ, VB, PJB, NB, RC, RD, JD, VGC, EG, DH, CH, JH, JLAK, SK, MJK, AL, EM, MM, AMM, KN, LO, SP, LR, KR, TR, CS, JS, ST, JW, NIW; resources, ASW, JPZ, VB, RC, JF, EG, DH, CJH, SK, JL, EM, AMM, LO, AIP, LR, TR, CS, JS, JT, JW; data curation, ASW; writing—original draft preparation, ASW, SMW, NMS, SPH; writing—review and editing, all co-

authors; visualization, ASW; supervision, ASW, JPZ, AIP, SK, JL, JS; project administration, ASW, NMS; funding acquisition, ASW, JPZ, RC, JF, EH, SK, JL, EM, AMM. The authors report no conflicts of interest.

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
