# Peer review of "Spatial and temporal variation in river corridor exchange across a 5th order mountain stream network"

_Hydrology and Earth System Sciences, 2019_

## Referee Comment (RC1) · Matt Cohen (Referee) · 3 Jun 2019

Adam and colleagues have developed a truly impressive data set from which they test a specific hypothesis about scaling of river corridor exchange. The topic is important, not least because it challenges some of the major pronouncements derived from steady-state models that assume network scaling rules. The technical treatment of the breakthrough curves is exemplary, spanning the full complement of modern approaches, and the writing is uniformly clear and compelling. In short, this is a paper that clearly merits publication. Below I document several areas where I found the paper in need of clarity, with one area in particular inviting at least some additional discussion if
not some new analysis (#2 below). My recommendation of minor revision is predicated on the former (discussion), recognizing that some additional statistical treatment of the responses would more accurately considered major revisions. I've also created a list of minor comments, provided in no particular order (typos, questions, comments).

1) Among the many technical strengths of this paper is the breadth of response metrics interpreting solute breakthrough curves. It is truly a smorgasbord of measures, consistent with the assembly of masters that comprise the author list. After a while, however, it ceased to be clear to me why so many metrics were necessary. The hypothesis is about predicting river corridor exchange with discharge, and while I would admit (and their results confirm) that we probably lack a singular measure of that exchange, the methods provided no specific rationale for the ones selected other than literature precedent, nor justify their independence from others selected. In figures 5 and 6, skewness finally emerged as the "response" and much of the paper would have been easier if the adequacy of this metric were proposed at the outset, justified theoretically, and supported empirically (e.g., as meaningfully covarying with other more complex response measures). Otherwise, despite an elegant hypothetico-deductive framework, the resulting effort feels a little like metric-fishing. I don't recommend removing metrics, but rather suggest making their selection strategic (rather than exhaustive) and supportive of general inference (rather than analyzed in parallel). And where that rationale is forced, then consider removing.

2) The setup for the research effort was exemplary. In the intro, the authors convey the existing conceptual model of river corridor exchange driven simultaneously by time- and space-varying discharge, as well as stream and valley geomorphic variation. A naïve view might be that these aspects act independently, but since changing discharge alters the head gradients that enable river-porewater exchange, and also the lateral and longitudinal geometry of the stream channel, the intro text points clearly to the plausibility (even primacy) of interactions. For this reason, the insistence on pairwise regression is confusing. There's a single passing acknowledgement (P20L21) that a
multivariate approach may be useful but no effort to explicitly consider contingency as a native feature of the question at hand. Framed as a question: is current theory consistent with interactions between geomorphology and discharge being important, or would such considerations be mostly a statistical contrivance? I believe it's the former, and that there's an opportunity with this data set to set the stage for future explorations of such interactions. If the authors agree, I think at least passing consideration of interaction terms is merited. If instead the authors feel conditional relationships are not implicitly supported by theory, say so explicitly. I'd note that the presentation of the Wondzell model in Fig. 6a implicitly suggests that the influences of watershed area and hyporheic potential are conditional (although in an additive sense); my contention is that there may indeed be informative interaction terms, and few data sets before this one are adequate to that challenge.

3) One core reason articulated (intro and discussion) for reduced river corridor exchange at high flow is that augmented hydraulic gradients to the stream compress hyporheic flowpaths. This is true when the hydraulic response in the stream and hillslope are synchronized. It seems demonstrably untrue otherwise, such as when flow generation is uneven (in small catchments) or when rainfall is uneven (in large catchments). Perhaps these are special cases, but the rivers where I've worked extensively exhibit significant "bank" storage during floods when storm-induced head changes are more rapid and pronounced in the stream than in the adjacent aquifer. The resulting hot moments of groundwater pumping into (and later out of) the hyporheic and bank sediments indicate that a simple monotonic association between instantaneous exchange and discharge is probably naïve. Only slightly less oversimplified might be to interrogate the river corridor exchange as a function of hydrograph position (or the time-rate of change of discharge) rather than discharge alone. I recommend the authors consider this. We did this for a setting where tidal variation created interesting hysteresis in hydraulic exchange (Hensley et al. 2015 WRR) and others (Audrey Sawyer among others) have seem similar dynamics. It's reasonable to rebut this comment by saying that explicit consideration of hydrograph position (or dQ/dt) invites an entirely different

paper, but the general critique of steady-state assumptions that underlies this work might be bolstered by avoiding the view of variable stream discharge as a sequence of steady-states. It is not.

Minor Comments: - P1L43. Should be "is" not "are". Or "exchange" should be "exchanges" - P2L4. The inclusion of the "and" between #2 and #3 underscores the interaction effects that may exist. - What does it mean (P6L19) for streams to change on annual to subannual time scales? Doesn't everything that changes at any time scale vary at all time scales? Do you mean that the streams change quickly? - I don't understand the rationale for stratifying by stream order (P7L5); more precisely, I don't understand why it was advantageous to bias the sampling to headwater sites over higher order reaches. The point here is not to characterize the network (where we might expect most of the variation to occur in the low order streams), but rather to explore geomorphic vs. discharge controls on river corridor exchange. To that end, a more balanced portfolio of sites makes more sense. I'll note that the resulting sample population (Fig. 3c) is pretty impressively distributed so this comment is more conceptual than operational. - It's been a while since I took a groundwater class, but why is the porosity term in the subsurface flow equation (P8)? Darcy's Law applies to the bulk cross section (here valley width times mean colluvium depth) and the Hvorslev K is for porous media. - I really appreciate the guidance on standardizing the reach length by wetted widths. I think this is an important standard operating procedure. - P9 refers to a companion manuscript. What/where is that? - The equations on P10L7-8 appear to have a typo. Doesn't the comparison for the conditional value have to be between CADE and COBS? I am confused how it could be CAD. - I really like the fMTS metric. It would be informative to consider how this compares with H (which I like less because I'm too dense to really understand it) and skewness (which I like a lot as well). For what it's worth, it was upon introduction of holdback (H) that the array of metrics started to seem excessive (or at least poorly defended). Some correlation among metrics (e.g., as a supplemental table) would be helpful. - For the SAS analysis, I was impressed by the explanation and by the utility of the metrics. I'd only note that the discharge

used (to compare against storage) is only surface stream discharge. The subsurface discharge (downvalley groundwater flow) is not included, and the relative importance of this flow depends strongly on network position. - P20L5 should be "hold" - The criterion of statistical significance is, of course, defensible, but I don't find the associations compelling just because they meet the criterion of being non-zero. The authors aren't trying to hide behind statistical significance, but seeing Table 2 made me wonder if the real story of these data (namely that we are really very poor at prediction of the thing we care most about) aren't a little too softened by putting pluses and minuses in almost every box. - On the subject of Table 2, I wonder if the predicted sign might be included somehow. For example, I would have (admittedly naively) predicted that skewness is reduced with increasing Q, UAA, V, order, width, and stream power, but perhaps not sinuosity or K. - QHEF on page 23 has the "HEF" subscripted. Elsewhere it's just "QHEF". - It's a little incongruous to show the overarching concept (Fig. 5) using watershed area and hyporheic potential, but then only use discharge for the pairwise plots. They are (Fig. 3a) clearly correlated, but not perfectly so. - Among the most important points is P31L10-12. We are mostly measuring in-stream storage with these short-term pulse tests. Unless we suppose that these high turnover storages are where most of the reactivity occurs (and I don't believe they are), efforts to link pulse-based breakthrough curves in a reach to network scale retention seems doomed to failure. The inclusion of metrics of storage proportion labelled by tracer is really informative.
* * *

---

## Referee Comment (RC2) · Anonymous Referee #2 · 18 Jun 2019

The work presented by Ward et al. represents an incredible amount of analysis based on an extensive dataset presented in a companion article. I was very excited to read and review this paper and hope that my comments will help improve it. The companion piece lays out data from synoptic and baseflow sampling of fluid fluxes through a variety of low order streams and this paper describes the analyses the team took to understand how exchange varies in relation to streamflow in space and time. With these analyses they seek to in/validate the model set forth by Wondzell (2011) and show that exchange decreases with increasing discharge through space, but that exchange varies in response with time in fixed stream reaches. Ward suggests a number of best practices for future large-scale sampling excursions to improve on these find-

ings and reach a more parsimonious conclusion—first, control for advective time; second, control for storage volume. Finally, they note that a multivariate approach is likely necessary to improve the systematic understanding of exchange in response to spatiotemporal variations in stream discharge. This is an important contribution to the discipline, and I will be delighted to see it in print after some revisions.

The introductory section argues convincingly that many parameters affect the exchange in streams – channel width, K, hydraulic gradient, etc. The authors spend a lot of time walking us through the measurement and calculation of many of these values, and some discussion of what those values mean and why they do or do not correlate with exchange. While this discussion is useful, I had trouble following all of the methods, results, and discussion. I think discussion of these parameters could be streamlined somewhat. For instance, I'm not sure that all of the panels of tables 3 and 4 belong in the body of this paper—several are not discussed and could be moved to the supplement. Additionally I spent a lot of time searching through the text to remind myself how each variable was defined. I think extra care could be taken when terms are defined, but I think most readers would find a list or table of variable definitions to be especially helpful.

In the results and discussion sections there is a brief mention that a multivariate approach is likely necessary to understand these relationships more thoroughly, but no analyses to investigate and present any such multivariate relationships. The authors return to this topic in the conclusion and argue that future studies must focus on these higher-level statistics. I would suggest the authors pursue this topic further within or at least explicitly discuss why they did not pursue this approach further. Ultimately, the authors reach the conclusion that skewness is the most predictive statistic. I think it is important to expand and further justify this conclusion—especially to explore a rationale for why skewness is a good indicator. I think it is also important to better support their claim with regards to skewness. In particular, I had trouble understanding figures 5 and 6. Figure 5 was of low image quality, so an enhanced resolution image might

have helped, but I had trouble seeing where the points were plotted in 3d space, and thus could not follow their argument. I found figure 6 unconvincing. The argument rests on best fit lines that don't seem supported by the underlying data. I would suggest replacing the figure, removing the lines, or at least presenting some statistical treatment of why they believe the best fit lines are justified.

A last concern is the number of authors—I am not used to seeing such a large author list on a data analysis paper. I think it is important to justify and define the contribution of each author toward the different tenets of authorship in a systematic manner—I think it is important that the authors make an earnest attempt to do so. One approach would be the approach suggested by Clement (2014).

Minor/general comments follow and are ordered chronologically. Pp:line:comment General: The paper would benefit greatly from a table/list of all variables at the start/end/supplement. I spent a lot of time flipping through the paper trying to remember what the variables and subscripts represented. 2:5: The "more than 60 solute tracer studies" were conducted in a companion paper, not this article, it is probably worth clarifying here and elsewhere. Careful throughout that data from the companion paper are not presented as results of this paper. 3: 13-14: is it expected that exchange volume will decrease or the ratio of $Q_{ex}/Q$? 3: 25: is it expected that exchange volume will decrease or the ratio of $Q_{ex}/Q$? Please clarify here and several other places. 6: Table 1: I suggest you change the order of table items to match order they're presented in the text. 7: 15-25: The presented replicate falling head tests were all conducted at one location in the stream channel. Were tests conducted to understand the spatial variability of K within the channel and floodplain sediments? K varies widely over relatively short scales, is there any way to bracket the errors associated with this? 7: 25: K is typically log-normal, should this be the log-geometric mean? 8: 4-7: If $Q_{sub,cap}$ is volumetric and based on Darcy, I don't understand why porosity is included in the calculation of the "capacity of the subsurface to convey water down to the valley bottom" as porosity should impact velocity only, and not impact volumetric flux. If porosity is

estimated as 30% for all sites, this shouldn't impact findings, but clarification would be helpful. 8:6-7: You say, "hvalley is the valley colluvium depth (m; estimated as 50% of the wetted channel width)". To clarify, depth of colluvium is never independently determined, it's only estimated as $\frac{1}{2}$ wetted channel width? If so, wetted at what stage (e.g. high discharge, mean discharge)? Please provide some references to support this as a valid approach. 8:9: Suggest changing "nor" to "or" 8:20-29: Please define more thoroughly the term "mixing length." Is this the length required for advective mixing to result in a homogeneous surface water concentration of a released solute? How was this determined in cases without any tracer. 9:1-2: The term "conflicting research" is unclear. Do you mean that you could not complete the test because other experiments meant that you could not do your own experiment, or that the findings of other experiments convinced you that your results were invalid, or something else? 10:4: Please clarify how MREC was determined. Is "mass recovered" the total mass recovered during the entire tracer test, the tracer test up to time t, or the mass recovered during the current time step? Also, how was a tracer test duration determined—was it continued until 99% recovery or something similar? 10:8-9: I'm confused about this equation. CAD (left hand side) is based on CAD (right hand side), which suggests CAD is known a priori? Should the RHS be CADE? 10:10: "associated with" is confusing. Do you mean something more like the "total solute mass" moved downstream by advection and dispersion? 10:20: same comment as above about "Associated with" Pp 10 and 11: 't' appears in some equation but not others that I expect to see it in. For instance, in all terms of "CTS=Cobs-CAD" I would expect the concentration to be a function of time. 11:3: Why 99% Is there some particular justification? Were you calculating this in the field to determine the length of time that tracer tests should be run? 13:7: What is tau? 13:15: What is "P"? Should this be "PQ"? I never see "P" defined. This is one of many cases where a symbology sheet would help immensely. 13:30: Again, what is "P"? 16:10: You never define the subscript "ds" in Cobs,ds so far as I can tell, thought you do define QDS. Please make sure all symbology is explicitly defined to remove confusion. Also, should this be "Cobs,DS" with the DS capitalized to match other us-

age? Pp 17: No reference to figure 3H, 3G is out of order. Fig 3. This symbology is difficult to interpret. I cannot distinguish symbology for the 4 streams from one another because the blues and greens are too similar, especially with the poor-resolution image of the submitted pdf. I suggest making all points translucent and making the colors of the non-synoptic samples more dissimilar. Also, I would recommend adding a curly bracket around the non-synoptic samples in the legend and labeling them as the stream-reach samples. The caption begins "for synoptic data" – please clarify caption to make it clear that the figures also include the non-synoptic data. Also clarify whether the line of "best fit" is for all data in panel or only for the synoptic data. Figure 4: Same comments as in figure 3. 20:4: "Hod" $\Rightarrow$ "Hold" 20:13: you say "most previous studies" but only cite one study. Please add more citations or remove statement. 20:22: You spent a lot of time showing and describing univariate values, but then say a multivariate approach is necessary to make sense of this data. Did you consider including some multivariate stats to explore these relationships? 22:27: "Sens slope was larger for the fixed reaches..." I don't recall if this is explicitly discussed later. 23:10-13: Is the decreased QHEF a volumetric decrease or a relative decrease as a fraction of stream discharge? Fig 5: What are the vertical columns? The colored lines? The right-hand panel is very difficult to interpret. The lefthand panel benefits from the lines that extend to z=0, to show the footprint of each point, whereas I cannot tell where points in the righthand panel exist in XY space. Is there a better way to present this data? Same comment about the color scheme as in figure 3 and 4—I cannot differentiate between the points. 28:21-2:Was this multi-sensor approach described in the methods of this paper? I did not see any previous mention. 28:25: Where were these data/results presented/discussed? I did not see previous mention in this paper. Fig 6: I do not trust the lines on these plots—I believe they are misleading and suggest they be removed. 31:10: Suggest removing "likely"

References: Clement, Prabhakar (2014). Authorship Matrix: A Rational Approach to Quantify Individual Contributions and Responsibilities in Multi-Author Scientific Articles. Sci Eng Ethics 20. 345–361. DOI 10.1007/s11948-013-9454-3

---

## Author Response (AR1)

Referees' comments in bold type. Authors responses below each comment.

Matt Cohen (Referee #1)
mjc@ufl.edu

**Adam and colleagues have developed a truly impressive data set from which they
test a specific hypothesis about scaling of river corridor exchange. The topic is important,
not least because it challenges some of the major pronouncements derived
from steady-state models that assume network scaling rules. The technical treatment
of the breakthrough curves is exemplary, spanning the full complement of modern
approaches, and the writing is uniformly clear and compelling. In short, this is a paper that
clearly merits publication. Below I document several areas where I found the paper in
need of clarity, with one area in particular inviting at least some additional discussion if
not some new analysis (#2 below). My recommendation of minor revision is predicated
on the former (discussion), recognizing that some additional statistical treatment of the
responses would more accurately considered major revisions. I've also created a list
of minor comments, provided in no particular order (typos, questions, comments).**

No response necessary as issues are detailed below.

**1) Among the many technical strengths of this paper is the breadth of response metrics
interpreting solute breakthrough curves. It is truly a smorgasbord of measures, consistent
with the assembly of masters that comprise the author list. After a while, however,
it ceased to be clear to me why so many metrics were necessary. The hypothesis is
about predicting river corridor exchange with discharge, and while I would admit (and
their results confirm) that we probably lack a singular measure of that exchange, the
methods provided no specific rationale for the ones selected other than literature
precedent, nor justify their independence from others selected. In figures 5 and 6, skewness
finally emerged as the "response" and much of the paper would have been easier if the
adequacy of this metric were proposed at the outset, justified theoretically, and supported
empirically (e.g., as meaningfully covarying with other more complex response
measures). Otherwise, despite an elegant hypothetico-deductive framework, the resulting
effort feels a little like metric-fishing. I don't recommend removing metrics, but
rather suggest making their selection strategic (rather than exhaustive) and supportive
of general inference (rather than analyzed in parallel). And where that rationale is
forced, then consider removing.**

Accepted. We have restructured the methods section to now include a discussion of why
these multiple approaches were implemented. Perhaps of most utility to the community,
we have added a new table (Table 2 in the revised manuscript) that details the relative
strengths and weaknesses of each approach. This table also include a summary of the key

metrics that are interpreted from each approach. Finally, while the list of metrics presented here is large, we consider it far from exhaustive.

Importantly, we do not intend this manuscript to indicate that skewness is somehow the most important or otherwise "best" metric to describe river corridor exchange. We intended it as illustrative of patterns that were consistent across many metrics. We have modified Figures 5 and 6 to now include multiple response variables to decrease the emphasis on skewness. This change, combined with the modified section 2.2 and newly added Table 2 should clarify this for readers.

2) The setup for the research effort was exemplary. In the intro, the authors convey the existing conceptual model of river corridor exchange driven simultaneously by time and space-varying discharge, as well as stream and valley geomorphic variation. A naïve view might be that these aspects act independently, but since changing discharge alters the head gradients that enable river-porewater exchange, and also the lateral and longitudinal geometry of the stream channel, the intro text points clearly to the plausibility (even primacy) of interactions. For this reason, the insistence on pairwise regression is confusing. There's a single passing acknowledgement (P20, L21) that a multivariate approach may be useful but no effort to explicitly consider contingency as a native feature of the question at hand. Framed as a question: is current theory consistent with interactions between geomorphology and discharge being important, or would such considerations be mostly a statistical contrivance? I believe it's the former, and that there's an opportunity with this data set to set the stage for future explorations of such interactions. If the authors agree, I think at least passing consideration of interaction terms is merited. If instead the authors feel conditional relationships are not implicitly supported by theory, say so explicitly. I'd note that the presentation of the Wondzell model in Fig. 6a implicitly suggests that the influences of watershed area and hyporheic potential are conditional (although in an additive sense); my contention is that there may indeed be informative interaction terms, and few data sets before this one are adequate to that challenge.

Accepted. Our focus here was testing Wondzell's (2011) conceptual model, not conducting a robust multivariate assessment nor exploring interactions between geologic setting and hydrologic forcing as controls. While we do find merit in this, and we indeed believe this data set is one of the first that might support this effort, it is beyond the scope of our study. That said, we have revised the manuscript to clarify that we did fit simple multivariate relationships to each response metric considered (i.e., the planar surface shown in Fig. 5 of the original study). We now describe this in the methods section and show multiple fits in the revised figures, plus include a comparison of univariate and bivariate fits in a supplemental table.

**3) One core reason articulated (intro and discussion) for reduced river corridor exchange at high flow is that augmented hydraulic gradients to the stream compress hyporheic flowpaths. This is true when the hydraulic response in the stream and hillslope are synchronized. It seems demonstrably untrue otherwise, such as when flow generation is uneven (in small catchments) or when rainfall is uneven (in large catchments).**

**Perhaps these are special cases, but the rivers where I've worked extensively exhibit significant "bank" storage during floods when storm-induced head changes are more rapid and pronounced in the stream than in the adjacent aquifer. The resulting hot moments of groundwater pumping into (and later out of) the hyporheic and bank sediments indicate that a simple monotonic association between instantaneous exchange and discharge is probably naïve. Only slightly less oversimplified might be to interrogate the river corridor exchange as a function of hydrograph position (or the time-rate of change of discharge) rather than discharge alone. I recommend the authors consider this. We did this for a setting where tidal variation created interesting hysteresis in hydraulic exchange (Hensley et al. 2015 WRR) and others (Audrey Sawyer among others) have seem similar dynamics. It's reasonable to rebut this comment by saying that explicit consideration of hydrograph position (or dQ/dt) invites an entirely different paper, but the general critique of steady-state assumptions that underlies this work might be bolstered by avoiding the view of variable stream discharge as a sequence of steady-states. It is not.**

> Accepted. We have clarified that Wondzell (2011) focused on differences in steady-state discharge by modifying the introduction and discussion, which is our focus in this study. This was stated in the last paragraph of the introduction: "variation in discharge as a function of drainage area during a fixed baseflow condition", but could have been more clear throughout the manuscript. Edits in response to this comment are primarily in describing Wondzell's (2011) discharge axis as "steady-state discharge" or "baseflow". We also added the following text to the introduction to differentiate steady-state differences from unsteady (i.e., dQ/dt, or hydrograph position) studies: "Notably, most classical expectations are based on differing steady discharge conditions (e.g., high vs. low baseflow), though an emerging body of field studies (detailed above), modeling studies (e.g., Malzone et al., 2016; Schmadel et al., 2016b), and conceptual models (e.g., Fig. 8 in Ward et al., 2016) are beginning to actively address exchange during unsteady discharge conditions."

**Minor Comments: -**
**P1L43. Should be "is" not "are". Or "exchange" should be "exchanges"**

> Accepted. Modified as suggested.

-
**P2L4. The inclusion of the "and" between #2 and #3 underscores the interaction effects that may exist. –**

> Accepted. Point taken, Dr. Cohn (no direct edit required in response to this comment)

**What does it mean (P6L19) for streams to change on annual to subannual time scales? Doesn't everything that changes at any time scale vary at all time scales? Do you mean that the streams change quickly? –**

> Accepted. We have removed the text "on annual to subannual timescales"

**I don't understand the rationale for stratifying by stream order (P7L5); more precisely, I don't understand why it was advantageous to bias the sampling to headwater sites over higher order reaches. The point here is not to characterize the network (where we might expect most of the variation to occur in the low order streams), but rather to explore geomorphic vs. discharge controls on river corridor exchange. To that end, a more balanced portfolio of sites makes more sense. I'll note that the resulting sample population (Fig. 3c) is pretty impressively distributed so this comment is more conceptual than operational. –**

Acknowledged. Site selection stratification was an attempt to meet multiple objectives of the field campaign, which are described in a high level in the related ESSD manuscript. In short, the overarching objective of the campaign itself was, indeed, to characterize the network. Thus, the desire for added samples in lower order streams where you correctly note we would expect more variation. The network-scale patterns presented in this study take the data as opportunistic, as we did not execute a separate campaign solely for this publication. However, we do note this is precisely one of the use-cases that we hoped for with the ESSD data – a community resource with sufficient sampling that it could be used to support any number of questions. No modifications to the study were made in response to this comment.

**It's been a while since I took a groundwater class, but why is the porosity term in the subsurface flow equation (P8)? Darcy's Law applies to the bulk cross section (here valley width times mean colluvium depth) and the Hvorslev K is for porous media. –**

Accepted. Nice catch! The porosity term here was a typo. We confirmed that in the data analysis the porosity was not used, and have corrected the equation accordingly.

**I really appreciate the guidance on standardizing the reach length by wetted widths. I think this is an important standard operating procedure. –**

Thanks!

**P9 refers to a companion manuscript. What/where is that? –**

Accepted. This refers to the paired submittal in Earth Systems Science Data. We have added the full citation to the "ESSD-D" paper in this location.

**The equations on P10L7-8 appear to have a typo. Doesn't the comparison for the conditional value have to be between CADE and COBS? I am confused how it could be CAD. –**

Acknowledged. We have confirmed that this formulation is correct and consistent with Wlostowski et al. (2017) where the approach is first published.

I really like the fMTS metric. It would be informative to consider how this compares with H (which I like less because I'm too dense to really understand it) and skewness (which I like a lot as well). For what it's worth, it was upon introduction of holdback (H) that the array of metrics started to seem excessive (or at least poorly defended). Some correlation among metrics (e.g., as a supplemental table) would be helpful. –

> Accepted. We have added a supplement to the manuscript that includes both tabular and visual representations of Pearson correlation and Spearman Rank Correlation.

**For the SAS analysis, I was impressed by the explanation and by the utility of the metrics. I'd only note that the discharge used (to compare against storage) is only surface stream discharge. The subsurface discharge (downvalley groundwater flow) is not included, and the relative importance of this flow depends strongly on network position. –**

> Accepted. We have revisited our analysis and confirmed that the denominator of the equation in question, Q (Page 14, Line 18) was used as the total down-valley discharge, not only the surface discharge. We have updated the denominator to now read "$Q+Q_{sub,cap}$" to clarify this point.

**P20L5 should be "hold" –**

> Accepted. Modified as suggested.

**The criterion of statistical significance is, of course, defensible, but I don't find the associations compelling just because they meet the criterion of being non-zero. The authors aren't trying to hide behind statistical significance, but seeing Table 2 made me wonder if the real story of these data (namely that we are really very poor at prediction of the thing we care most about) aren't a little too softened by putting pluses and minuses in almost every box. –**

> Accepted. We wholeheartedly agree! Indeed, the reason we included r2 in the table was to demonstrate that while we may find trends that pass a statistical test making them likely to have one direction or the other, these offer very little predictive power. We discuss this in the paragraph that preceded Table 2 (page 20 lines 15+ in the original submittal). Our confidence bolstered by this comment, we have added the following text to the conclusions to underscore this point: "Importantly, we document consistent trends with discharge that have low explanatory power (low $r^2$) despite being statistically significant in their direction, indictaing that we have little predictive power"

**On the subject of Table 2, I wonder if the predicted sign might be included somehow. For example, I would have (admittedly naively) predicted that skewness is reduced with increasing Q, UAA, V, order, width, and stream power, but perhaps not sinuosity or K. –**

> Acknowledged. We agree with this idea, in concept, but do not believe that there exists a consistent expectation for each of these metrics. Ward and Packman (2018) document

that conflicting predictions exist for nearly any outcome of interest (exchange flux, timescale, hyporheic geometry) as a function of any geologic or hydrologic input.

Ward AS, Packman AI. (2018). Advancing our predictive understanding of river corridor exchange. WIREs Water . 2018;e1327. https://doi.org/10.1002/wat2.1327

**QHEF on page 23 has the "HEF" subscripted. Elsewhere it's just "QHEF". –**

Accepted. Modified to use the subscript throughout.

**It's a little incongruous to show the overarching concept (Fig. 5) using watershed area and hyporheic potential, but then only use discharge for the pairwise plots. They are (Fig. 3a) clearly correlated, but not perfectly so. –**

Accepted. We have added a supplement showing the Pearson and Spearman's Rank correlations between all pairs of site descriptors and metrics, including both tabular data and a visualization. We have also included versions of Fig. 3 and Fig. 4 that include HYPPOT and UAA on the X-axis as these are the variables used by Wondzell (2011).

**Among the most important points is P31L10-12. We are mostly measuring in-stream storage with these short-term pulse tests. Unless we suppose that these high turnover storages are where most of the reactivity occurs (and I don't believe they are), efforts to link pulse-based breakthrough curves in a reach to network scale retention seems doomed to failure. The inclusion of metrics of storage proportion labelled by tracer is really informative.**

Thanks!

**Anonymous Referee #2**

**The work presented by Ward et al. represents an incredible amount of analysis based on an extensive dataset presented in a companion article. I was very excited to read and review this paper and hope that my comments will help improve it. The companion piece lays out data from synoptic and baseflow sampling of fluid fluxes through a variety of low order streams and this paper describes the analyses the team took to understand how exchange varies in relation to streamflow in space and time. With these analyses they seek to in/validate the model set forth by Wondzell (2011) and show that exchange decreases with increasing discharge through space, but that exchange varies in response with time in fixed stream reaches. Ward suggests a number of best practices for future large-scale sampling excursions to improve on these findings and reach a more parsimonious conclusãˇAˇ Tfirst, control for advective time; second, control for storage volume. Finally, they note that a multivariate approach is likely necessary to improve the systematic understanding of exchange in response to spatiotemporal variations in stream discharge. This is an important contribution to the discipline, and I will be delighted to see it in print after some revisions.**

No responses necessary to the comment above, as issues are addressed in more detail below.

**The introductory section argues convincingly that many parameters affect the exchange in streams – channel width, K, hydraulic gradient, etc. The authors spend a lot of time walking us through the measurement and calculation of many of these values, and some discussion of what those values mean and why they do or do not correlate with exchange. While this discussion is useful, I had trouble following all of the methods, results, and discussion. I think discussion of these parameters could be streamlined somewhat. For instance, I'm not sure that all of the panels of tables 3 and 4 belong in the body of this paperăˇAˇTseveral are not discussed and could be moved to the supplement. Additionally I spent a lot of time searching through the text to remind myself how each variable was defined. I think extra care could be taken when terms are defined, but I think most readers would find a list or table of variable definitions to be especially helpful.**

> Accepted. We have added a table to the manuscript that summarizes the various approaches and key metrics (Table 2 in the revised study). However, we have elected to keep the metrics all in the study for sake of completeness, and because we do not believe any of them to be redundant.

**In the results and discussion sections there is a brief mention that a multivariate approach is likely necessary to understand these relationships more thoroughly, but no analyses to investigate and present any such multivariate relationships. The authors return to this topic in the conclusion and argue that future studies must focus on these higher-level statistics. I would suggest the authors pursue this topic further within or at least explicitly discuss why they did not pursue this approach further.**

> See response to major comment #2 for the first referee.

**Ultimately, the authors reach the conclusion that skewness is the most predictive statistic. I think it is important to expand and further justify this conclusionăˇAˇTespecially to explore a rationale for why skewness is a good indicator. I think it is also important to better support their claim with regards to skewness.**

> Acknowledged. We disagree with the reviewers statement "Ultimately, the authors reach the conclusion that skewness is the most predictive statistic." We selected skewness as a representative and easily understood variable to demonstrate our point in Figs. 5-6, but do not consider it to be a singular "Best" variable. We have taken care to clarify this by adding other metrics to Fig. 5 and Fig. 6 and emphasized this in a brief discussion of how metrics were selected (first paragraph in section 2.2 in the revised manuscript).

**In particular, I had trouble understanding figures 5 and 6. Figure 5 was of low image quality, so an enhanced resolution image might have helped, but I had trouble seeing where the points were plotted in 3d space, and thus could not follow their argument. I found figure 6 unconvincing. The argument rests on best fit lines that don't seem supported by**

**the underlying data. I would suggest replacing the figure, removing the lines, or at least presenting some statistical treatment of why they believe the best fit lines are justified.**

Accepted. Figure 5 in the original version (Fig. 6 in the revised) has been revised to improve the visualization and interpretability of the data. Figure 6 in the original study has been moved to Fig. 5 in the revised form. The figure now depicts t99, holdback, and skewness as a function of advective time. The revised figure is described in the results (newly added section 3.3). We retain the linear trend-lines as a useful interpretive tool, but provide a quantitative comparison of the ranges of parameter values between the different approaches.

**A last concern is the number of authorsâ˘Aˇ TI am not used to seeing such a large author list on a data analysis paper. I think it is important to justify and define the contribution of each author toward the different tenets of authorship in a systematic mannerâ˘Aˇ TI think it is important that the authors make an earnest attempt to do so. One approach would be the approach suggested by Clement (2014).**

Acknowledged. We respectfully note that author contributions were described, albeit briefly, in the acknowledgements section of the manuscript. The lead author hereby confirms that each co-author contributed at a level consistent with Clement's (2014) recommendations. This is perhaps best understood by the scope of the field campaign that was required to characterize these sites, the many approaches taken to interpret the data, a collaborative writing process where all co-authors were active participants, and a team that has been working together for several years on a series of collaborative projects.

**Minor/general comments follow and are ordered chronologically.**
**Pp:line:comment**
**General: The paper would benefit greatly from a table/list of all variables at the start/end/supplement. I spent a lot of time flipping through the paper trying to remember what the variables and subscripts represented.**

Accepted. The newly added Table 2 include a summary of the key response variables that are used in this study.

**2:5: The "more than 60 solute tracer studies" were conducted in a companion paper, not this article, it is probably worth clarifying here and elsewhere. Careful throughout that data from the companion paper are not presented as results of this paper.**

Accepted. We have modified this sentence to now read: "To test this conceptual model we conducted more than 60 solute tracer studies including a synoptic campaign in the 5th order river network of the H.J. Andrews Experimental Forest (Oregon, USA) and replicate-in-time experiments in four watersheds.". We have elected not to include a reference to the ESSD companion paper in the abstract, but make clear reference to this data set later in the study.

**3: 13-14: is it expected that exchange volume will decrease or the ratio of Qex/Q?**

**3: 25: is it expected that exchange volume will decrease or the ratio of Qex/Q? Please clarify here and several other places.**

> Acknowledged. There is not a predominant expectation of this relationship. One could argue that if $Q_{HEF}$ is constant (for example, due to some geologic feature that controls exchange flux and does not change with discharge), then increasing discharge would decrease $Q_{HEF}/Q$. However, other mechanisms (e.g., diffusion of turbulent momentum across the streambed) may vary with discharge, changing both $Q_{HEF}$ and $Q$ simultaneously.

**6:Table 1: I suggest you change the order of table items to match order they're presented in the text.**

> Accepted. Table order has been modified as suggested.

**7: 15-25: The presented replicate falling head tests were all conducted at one location in the stream channel. Were tests conducted to understand the spatial variability of K within the channel and floodplain sediments? K varies widely over relatively short scales, is there any way to bracket the errors associated with this?**
**7: 25: K is typically log-normal, should this be the log-geometric mean?**

> Acknowledged. The tests to estimate K were conducted at a single location at each study site. The value reported is the geometric mean, taken from the published data set detailed in the ESSD manuscript.

**8: 4-7: If Qsub,cap is volumetric and based on Darcy, I don't understand why porosity is included in the calculation of the "capacity of the subsurface to convey water down to the valley bottom" as porosity should impact velocity only, and not impact volumetric flux. If porosity is estimated as 30% for all sites, this shouldn't impact findings, but clarification would be helpful.**

> Accepted. Porosity was included in the equation as a typo. We confirmed it was not used in the calculations, and this reviewer is correct that it should not have been there. The equation and text have been updated accordingly.

**8:6-7: You say, "hvalley is the valley colluvium depth (m; estimated as 50% of the wetted channel width)". To clarify, depth of colluvium is never independently determined, it's only estimated as 1/2 wetted channel width? If so, wetted at what stage (e.g. high discharge, mean discharge)? Please provide some references to support this as a valid approach.**

> Accepted. We have added several references here that have estimated depth of colluvium for several sites in the study basin. We have added the following text: "This estimate is consistent with depths used in past studies (Gooseff et al., 2006; Ward et al., 2012; Crook

et al., 2008; Ward et al., 2018a; 2018c; Schmadel et al., 2017 ) and geophysical transects in the 4[th] and 5[th] order reaches of Lookout Creek (Wondzell, unpublished data)."

**8:9: Suggest changing "nor" to "or"**

Accepted. Modified as suggested.

**8:20-29: Please define more thoroughly the term "mixing length." Is this the length required for advective mixing to result in a homogeneous surface water concentration of a released solute? How was this determined in cases without any tracer.**

We have added the following text to clearly define the term "mixing length" at its first use in the manuscript: "(i.e., the distance required for the solute tracer to be well-mixed across the channel cross-section)". We have also added the following text to describe how mixing length was estimated in the field: "Mixing lengths were based on visual estimates in the field as empirical estimates are unreliable in mountain streams (Day et al., 1977). Moreover, field experience in a study system is recognized to be potentially more useful that theoretical estimates of mixing length (Kilbatrick and Cobb, 1985). Thus, we used visual estimates that are consistent with our past studies using these techniques and tracers in H.J. Andrews Experimental Forest (Ward et al., 2012; 2013a; 2013b; 2019; Voltz et al., 2013) and practices used in other mountain stream networks (e.g., Payn et al., 2009; Covino et al., 2010)."

**9:1-2: The term "conflicting research" is unclear. Do you mean that you could not complete the test because other experiments meant that you could not do your own experiment, or that the findings of other experiments convinced you that your results were invalid, or something else?**

Accepted. The sentence has been modified to more clearly explain, now reading "The differing number of replicates reflects either sensor failure or omission of a replicate due to conflicting research occurring at the same sites by other researchers (i.e., our replication would have negatively impacted their independent research campaigns, so we did not proceed with our injections)."

**10:4: Please clarify how MREC was determined. Is "mass recovered" the total mass recovered during the entire tracer test, the tracer test up to time t, or the mass recovered during the current time step? Also, how was a tracer test duration determined˜A ˘Twas it continued until 99% recovery or something similar?**

Accepted. MREC was previously defined in section 2.5. We have moved that definition up into section 2.2 where it is first used.

**10:8-9: I'm confused about this equation.**
**CAD (left hand side) is based on CAD (right hand side), which suggests CAD is known a priori? Should the RHS be CADE?**

Accepted. The right-hand side "CAD" should have been "CADE", which has now been corrected.

**10:10: "associated with" is confusing. Do you
mean something more like the "total solute mass" moved downstream by advection
and dispersion?**
**10:20: same comment as above about "Associated with" Pp 10 and**

Acknowledged. We have elected to retain this language as it is consistent with the original publication of these techniques (Wlostowski et al., 2017).

**10:11: 't' appears in some equation but not others that I expect to see it in. For instance,
in all terms of "CTS=Cobs-CAD" I would expect the concentration to be a function of
time.**

Accepted. Several equations were missing "(t)" in this section, all of which have been updated.

**11:3: Why 99% Is there some particular justification? Were you calculating this
in the field to determine the length of time that tracer tests should be run?**

Accepted. This truncation is performed post-hoc to minimize the disproportionally high impact of late-time noise on summary metrics calculated for short term storage, and is consistent with many past studies of solute tracer transport. We have added the following text to clarify this: "…consistent with common practices (e.g., Mason et al., 2012; Ward et al., 2013a; 2013b; Schmadel et al., 2016) and a community tool for interpretation of solute tracers (Ward et al., 2017a)."

**13:7: What is tau?**

Accepted. We have added the following text to define tau: "where $\tau$ is a random variable representing the age of a parcel of water (Harman, 2015)".

**13:15: What is "P"? Should this be "PQ"? I never see "P" defined. This is one
of many cases where a symbology sheet would help immensely.**
**13:30: Again, what is "P"?**

Accepted. In both cases, "$P$" has been replaced with "$P_Q$".

**16:10: You never define the subscript "ds" in Cobs,ds so far as I can tell, thought
you do define QDS. Please make sure all symbology is explicitly defined to remove
confusion. Also, should this be "Cobs,DS" with the DS capitalized to match other us-
age?**

Accepted. We have dropped the "ds" convention as Cobs is always used in reference to the downstream solute tracer timeseries. We have clearly defined Cobs where it is first

used in the study: "where $C_{obs,DS}$ (g m$^{-3}$) is the observed solute tracer concentration at the downstream location in response to the upstream solute tracer injection".

**Pp 17: No reference to figure 3H, 3G is out of order.**

Accepted. WE have added the reference to Fig. 3H in the paragraph. We have elected not to re-write the paragraph to address the order in which subplots are discussed.

**Fig 3. This symbology is difficult to interpret. I cannot distinguish symbology for the 4 streams from one another because the blues and greens are too similar, especially with the poor-resolution image of the submitted pdf. I suggest making all points translucent and making the colors of the non-synoptic samples more dissimilar. Also, I would recommend adding a curly bracket around the non-synoptic samples in the legend and labeling them as the stream-reach samples. The caption begins "for synoptic data" – please clarify caption to make it clear that the figures also include the non-synoptic data. Also clarify whether the line of "best fit" is for all data in panel or only for the synoptic data. Figure 4: Same comments as in figure 3.**

Acknowledged. We have used both shape and color to distinguish the sites, and have elected to retain this redundant differentiation to help readers. The colors are selected from the "Parula" colormap in Matlab, which is designed to retain contrast in greyscale and color prints and be accessible for color-impaired vision. We have clarified the symbology by adding the following text to the figure caption: "Data from unnamed creek (triangles, Cold creek (squares), WS03 (diamonds), and WS01 (stars) show the repeated injections through baseflow recession each headwater catchment."

**20:4: "Hod" ) "Hold"**

Accepted. Modified as suggested.

**20:13: you say "most previous studies" but only cite one study. Please add more citations or remove statement.**

Accepted. The one study cited is a notable exception to the "most" that we were referring to. We have modified the sentence to read: "Thus, while our selection of study reach lengths was imperfect to achieve identical advective timescales, we contend that we have adequately controlled for advective time."

**20:22: You spent a lot of time showing and describing univariate values, but then say a multivariate approach is necessary to make sense of this data. Did you consider including some multivariate stats to explore these relationships?**

Accepted. See response to Reviewer #1's "major comment 2"

**22:27: "Sens slope was larger for the fixed reaches: : :" I don't recall if this is explicitly discussed later.**

Acknowledged. This point is discussed again in the conclusions of the study.

**23:10-13: Is the decreased QHEF a volumetric decrease or a relative decrease as a fraction of stream discharge?**

Acknowledged. The text in question reads "We did find an increasing fraction of total discharge sampled in higher discharge locations (Fig. 4C), but the overall trend indicates that $Q_{HEF}$ does not grow as rapidly as $Q$, moving downstream along the network.". We believe this text is sufficiently clear in its current form as an interpretation of changes in QHEF, not to the quantity QHEF/Q.

**Fig 5: What are the vertical columns? The colored lines? The right-hand panel is very difficult to interpret. The lefthand panel benefits from the lines that extend to z=0, to show the footprint of each point, whereas I cannot tell where points in the righthand panel exist in XY space. Is there a better way to present this data? Same comment about the color scheme as in figure 3 and 4ãˇAˇ TI cannot differentiate between the points.**

Accepted. We have revised the figure to highlight only the synoptic data, added "stems" to orient the data in the bottom X-Y plane of the figure, and included colored best-fit planar surfaces to aid in visualization.

**28:21-2:Was this multi-sensor approach described in the methods of this paper? I did not see any previous mention.**

Accepted. We have moved the appropriate portions of the text from the locations referenced here to the methods (newly added section 2.1.4) and results (newly added section 3.3).

**28:25: Where were these data/results presented/discussed? I did not see previous mention in this paper. Fig 6: I do not trust the lines on these plotsãˇA ˇ TI believe they are misleading and suggest they be removed.**

See response to the third major comment from this reviewer.

**31:10: Suggest removing "likely"**

Accepted. Modified as suggested.

**References: Clement, Prabhakar (2014). Authorship Matrix: A Rational Approach to Quantify Individual Contributions and Responsibilities in Multi-Author Scientific Articles. Sci Eng Ethics 20. 345–361. DOI 10.1007/s11948-013-9454-3**

**Spatial and temporal variation in river corridor exchange across a 5th order mountain stream network**

Adam S. Ward[1], Steven M. Wondzell[2], Noah M. Schmadel[1,3], Skuyler Herzog[1], Jay P. Zarnetske[4], Viktor Baranov[5,6], Phillip J. Blaen[7,8,9], Nicolai Brekenfeld[7], Rosalie Chu[10], Romain Derelle[11], Jennifer Drummond[7,12], Jan Fleckenstein[13,14], Vanessa Garayburu-Caruso[15], Emily Graham[15], David Hannah[7], Ciaran Harman[16], Jase Hixson[1], Julia L.A. Knapp[17,18], Stefan Krause[7], Marie J. Kurz[13,19], Jörg Lewandowski[20,21], Angang Li[22], Eugènia Martí[12], Melinda Miller[1], Alexander M. Milner[7], Kerry Neil[1], Luisa Orsini[11], Aaron I. Packman[22], Stephen Plont[4,23], Lupita Renteria[24], Kevin Roche[25], Todd Royer[1], Catalina Segura[26], James Stegen[15], Jason Toyoda[10], Jacqueline Wells[24], Nathan I. Wisnoski[27]

[1] O'Neill School of Public and Environmental Affairs, Indiana University, Bloomington, Indiana, USA
[2] USDA Forest Service, Pacific Northwest Research Station, Corvallis, Oregon, USA.
[3] Earth Surface Processes Division, U.S. Geological Survey, Reston, Virginia, USA
[4] Department of Earth and Environmental Sciences, Michigan State University, East Lansing, Michigan, USA
[5] LMU Munich Biocenter, Department of Biology II, Großhaderner Str. 2, 82152 Planegg-Martinsried, Germany
[6] Department of River Ecology and Conservation, Senckenberg Research Institute and Natural History Museum, 63571 Gelnhausen, Germany
[7] School of Geography, Earth & Environmental Sciences, University of Birmingham, Edgbaston. Birmingham. B15 2TT. UK
[8] Birmingham Institute of Forest Research (BIFoR), University of Birmingham, Edgbaston. Birmingham. B15 2TT. UK
[9] Yorkshire Water, Halifax Road, Bradford, BD6 2SZ
[10] Environmental Molecular Sciences Laboratory, Pacific Northwest National Laboratory, Richland, WA, USA
[11] Environmental Genomics Group, School of Biosciences, the University of Birmingham, Birmingham B15 2TT, UK
[12] Integrative Freshwater Ecology Group, Centre for Advanced Studies of Blanes (CEAB-CSIC), Blanes, Spain
[13] Dept. of Hydrogeology, Helmholtz Center for Environmental Research - UFZ, Permoserstraße 15, 04318 Leipzig, Germany
[14] Bayreuth Center of Ecology and Environmental Research, University of Bayreuth, 95440 Bayreuth, Germany
[15] Earth and Biological Sciences Division, Pacific Northwest National Laboratory, Richland, WA, USA
[16] Department of Environmental Health and Engineering, Johns Hopkins University, Baltimore, Maryland, USA
[17] Department of Environmental Systems Science, ETH Zürich, Zurich, Switzerland
[18] Center for Applied Geoscience, University of Tübingen, Tübingen, Germany
[19] The Academy of Natural Sciences of Drexel University, Philadelphia, Pennsylvania, USA
[20] Leibniz-Institute of Freshwater Ecology and Inland Fisheries, Department Ecohydrology, Müggelseedamm 310, 12587 Berlin, Germany
[21] Humboldt University Berlin, Geography Department, Rudower Chaussee 16, 12489 Berlin, Germany
[22] Department of Civil and Environmental Engineering, Northwestern University, Evanston, Illinois, USA
[23] Department of Biological Sciences, Virginia Polytechnic Institute and State University, Blacksburg, Virginia, USA
[24] Pacific Northwest National Laboratory, Richland, WA, USA
[25] Department of Civil & Environmental Engineering & Earth Sciences, University of Notre Dame, Notre Dame, IN
[26] Forest Engineering, Resources, and Management, Oregon State University Corvallis, OR, USA
[27] Department of Biology, Indiana University, Bloomington, Indiana, USA

*Correspondence to*: Adam S. Ward (adamward@indiana.edu)

**Abstract.** Although most field and modeling studies of river corridor exchange have been conducted a scales ranging from 10's to 100's of meters; results of these studies are used to predict their ecological

*This draft manuscript is distributed solely for purposes of scientific peer review. Its content is deliberative and predecisional, so it must not be disclosed or released by reviewers. Because the manuscript has not yet been approved for publication by the U.S. Geological Survey (USGS), it does not represent any official USGS finding or policy.*

and hydrological influences at the scale of river networks. Further complicating prediction, exchanges are expected to vary with hydrologic forcing and the local geomorphic setting. While we desire predictive power, we lack a complete spatiotemporal relationship relating discharge to the variation in geologic setting and hydrologic forcing that are expected across a river basin. Indeed, Wondzell's
5   (2011) conceptual model predicts systematic variation in river corridor exchange as a function of (1) variation in baseflow over time at a fixed location, (2) variation in discharge with location in the river network, and (3) local geomorphic setting. To test this conceptual model we conducted more than 60 solute tracer studies including a synoptic campaign in the 5th order river network of the H.J. Andrews Experimental Forest (Oregon, USA) and replicate-in-time experiments in four watersheds. We interpret
10  the data using a series of metrics describing river corridor exchange and solute transport, testing for consistent direction and magnitude of relationships relating these metrics to discharge and local geomorphic setting. We confirmed systematic decrease in river corridor exchange space through the river networks, from headwaters to the larger mainstem. However, we did not find systematic variation with changes in discharge through time, nor with local geomorphic setting. While interpretation of our
15  results is complicated by problems with the analytical methods, they are sufficiently robust for us to conclude that space-for-time and time-for-space substitutions are not appropriate in our study system. Finally, we suggest two strategies that will improve the interpretability of tracer test results and help the hyporheic community develop robust data sets that will enable comparisons across multiple sites and/or discharge conditions.

20  **1 Introduction**

[revised manuscript text omitted]

**Commented [KTV9]:** In the text these sites are introduced from low elevation to high elevation (WS01, WS03, Unnamed Creek, Cold Creek), which is different from the order in this table. I suggest reordering this table to match the order in the text unless there are gradients in the controls that are important to represent in this table.

**Commented [n10R9]:** Thank you for the suggestion and detailed focus of the text. However, we feel that listing them as is in the table is more appropriate for a stand-alone table because WS01 and WS03 appear in many studies in the literature and will connect with broader readership, whereas Cold Cr. And Unnamed Creek are essential new watersheds in the present study. As described in the text, we feel is makes more sense to discuss these watersheds in order of gradients.

**Commented [KTV11]:** The formatting of in text citations switches to *italics* here, when they are not italicised above. Suggest adjusting for consistency.

**Commented [n12R11]:** Thank you for catching this inconsistency in formatting. We well correct all referencing for consistency prior to submitting a revised version to the journal.

[revised manuscript text omitted]

20    their independent research campaigns, so we did not proceed with our injections). These sites parallel the common approach of replication of a study at a fixed reach with varied discharge to relate river corridor exchange to discharge conditions (after Payn et al., 2009).

Commented [KTV17]: What equipment was used to monitor in-stream specific conductance? What is the expected level of accuracy with these sensors? Were multiple sensors deployed across the cross-section to estimate if completely mixed conditions were met?

Commented [n18R17]: This analysis paper was submitted as a companion data collection paper that contains more details, like you point out, regarding the data collection. However, we agree and have repeated some of the pertinent information here.

Commented [KTV19]: Do the authors have a sense if the sodium ions are as conservative in the study reaches as they are in the stream water matrix used to generate the standard curves?

My thought process here is that the tracer is exposed to a wider range of environmental conditions in the stream, including contact with substrate, than the tracer used to develop the standard curves.

As an end member, consider a case where all the sodium is removed from the stream via biogeochemical processes. In this case, the specific conductance signal produced from the chloride that was not lost would be lower than if the sodium were still present, and it would appear that a portion of the tracer (and therefore water) was lost in the study reach. This is clearly not the case, as evidenced by the chloride remaining in the water column, but it could result in an overestimation of $M_{rec}$ and the dependant equations.

Determining sodium loss is beyond the scope of this paper, and I suggest that this issue could be addressed with a discussion of the assumption of the conservative nature of both sodium and chloride across the study reaches.

Commented [n20R19]: Thank you for the comment. Yes, we assume that NaCl is conservative and added text to point that assumption out to the reader. Given the short travel times we do not suspect biogeochemical losses of Na. Also, to prevent introducing a bias in the observations due to assuming Na is also conservative, we build calibration curves using stream water.

Commented [KTV21]: Did these varying amounts cover the full range of specific conductance measured in the breakthrough curves?

Commented [n22R21]: Thanks for the important question. Yes, calibration curves were cover the full range of specific conductance measured in the field. There were no specific conductance breakthrough curves with values larger than that high end of the calibration curve.

While it is important to make sure measurements are within the range of the constructed calibration curve, we do not think adding this detail here is necessary as this method of using salt tracers is well established.

[revised manuscript text omitted]

Commented [KTV41]: It would be helpful to have a table of reach lengths and widths for each injection to illustrate this point. This table could go in an SI, as it would likely be quite large. If the authors decide to produce such a table, it would also be a good place to include NaCl injection masses, travel times, etc.

Commented [n42R41]: All the data used in this paper will be published and made publicly available, including all reach lengths, tracer masses, etc. We feel that including such a table as part of this manuscript may introduce confusion because we present metrics derived from that data that provide meaningful process understanding whereas lengths themselves do not.

**Table 3.** Mann-Kendall tests indicate significant ($p < 0.05$), monotonic trends relating almost all site characteristics and metrics of river corridor exchange for the synoptic survey locations. The direction of the trend is indicated as increasing ("+") or decreasing ("-"). Three relationships that lacked a significant trend are denoted "?" in the table below. Additionally, the magnitude of the coefficient of determination ($r^2$) for univariate power-law fit is presented as an indicator of the power of a simple regression.

| | Experimental Design | | | AD vs. TS | Short term storage | | | | rSAS | | | Long-term storage | | Metric |
|---|---|---|---|---|---|---|---|---|---|---|---|---|---|---|
| | $t_{99}$ | $t_{peak}$ | $L_{detect}$ | $f_{MAD}$ | $M_1$ | CV | v | Holdback | $f_{Qlabeled}$ | $f_{VSTR}$ | $f_{VTOT}$ | $f_{QGAIN,MAX}$ | $f_{QLOSS,MAX}$ | |
| Q | (-) | + | + | + | + | (-) | (-) | (-) | + | (-) | (-) | (-) | (-) | |
| UAA | + | + | + | (-) | + | (-) | (-) | (-) | (-) | (-) | (-) | + | (-) | |
| V | (-) | + | + | + | (-) | (-) | (-) | (-) | + | (-) | (-) | + | + | |
| Stream Order | + | + | ? | ? | + | (-) | (-) | (-) | (-) | (-) | (-) | + | (-) | Sen's Slope |
| K | (-) | + | (-) | + | + | (-) | (-) | (-) | (-) | (-) | (-) | + | (-) | |
| Down-Valley Slope | (-) | (-) | (-) | (-) | (-) | + | + | + | (-) | + | + | (-) | (-) | |
| Valley Width | + | + | + | + | + | (-) | (-) | (-) | + | (-) | (-) | (-) | (-) | |
| Stream Power | (-) | + | + | + | + | (-) | (-) | (-) | + | (-) | (-) | (-) | (-) | |
| $Q_{sub,cap}$ | (-) | + | (-) | + | + | (-) | (-) | (-) | (-) | (-) | (-) | + | (-) | |
| Sinuosity | + | + | (-) | + | + | + | (-) | + | ? | (-) | (-) | (-) | (-) | |
| Q | 0.07 | 0.00 | 0.37 | 0.00 | 0.04 | 0.05 | 0.15 | 0.02 | 0.00 | 0.15 | 0.00 | 0.92 | 0.23 | |
| UAA | 0.00 | 0.11 | 0.51 | 0.01 | 0.03 | 0.06 | 0.13 | 0.01 | 0.00 | 0.18 | 0.05 | 0.00 | 0.02 | |
| V | 0.18 | 0.00 | 0.29 | 0.02 | 0.08 | 0.08 | 0.03 | 0.01 | 0.02 | 0.19 | 0.01 | 0.01 | 0.00 | |
| Stream Order | 0.01 | 0.10 | 0.03 | 0.00 | 0.04 | 0.02 | 0.10 | 0.00 | 0.00 | 0.14 | 0.04 | 0.00 | 0.03 | |
| K | 0.02 | 0.03 | 0.10 | 0.02 | 0.00 | 0.13 | 0.10 | 0.04 | 0.01 | 0.07 | 0.01 | 0.03 | 0.11 | |
| Down-Valley Slope | 0.01 | 0.19 | 0.03 | 0.04 | 0.07 | 0.05 | 0.04 | 0.01 | 0.00 | 0.10 | 0.13 | 0.01 | 0.02 | $r^2$ for power law fit |
| Valley Width | 0.00 | 0.02 | 0.01 | 0.01 | 0.01 | 0.06 | 0.16 | 0.03 | 0.00 | 0.06 | 0.09 | 0.92 | 0.34 | |
| Stream Power | 0.07 | 0.00 | 0.74 | 0.00 | 0.04 | 0.05 | 0.15 | 0.02 | 0.00 | 0.15 | 0.00 | 0.92 | 0.23 | |
| $Q_{sub,cap}$ | 0.02 | 0.03 | 0.62 | 0.00 | 0.00 | 0.19 | 0.30 | 0.07 | 0.00 | 0.17 | 0.03 | 0.02 | 0.11 | |
| Sinuosity | 0.01 | 0.00 | 0.01 | 0.01 | 0.00 | 0.00 | 0.03 | 0.00 | 0.00 | 0.03 | 0.01 | 0.92 | 0.22 | |
| MAX | 0.18 | 0.19 | 0.74 | 0.04 | 0.08 | 0.19 | 0.30 | 0.07 | 0.02 | 0.19 | 0.13 | 0.92 | 0.34 | |
| MEDIAN | 0.02 | 0.03 | 0.20 | 0.01 | 0.04 | 0.05 | 0.11 | 0.02 | 0.00 | 0.14 | 0.02 | 0.02 | 0.11 | |
| MEAN | 0.04 | 0.05 | 0.27 | 0.01 | 0.03 | 0.07 | 0.12 | 0.02 | <0.01 | 0.12 | 0.04 | 0.37 | 0.13 | |
| MIN | <0.01 | <0.01 | 0.01 | <0.01 | <0.01 | <0.01 | 0.03 | <0.01 | <0.01 | 0.03 | <0.01 | <0.01 | <0.01 | |

**3.2.2 Fixed reach vs. synoptic results**

We found decreasing $t_{99}$ with increasing discharge for the synoptic study (Fig. 4D), which in turn resulted in a systematic reduction in the possible length of flowpaths that could be detected by tracer (Fig. 4G). Note that this ranges, on average, from 0.35 m at the lowest discharge to only 0.09 m at the highest discharge and the reach with the largest $L_{detect}$ was only 2.0 m. In contrast, reach lengths used in the fixed reach studies were much longer relative to stream size than the synoptic reaches, thus $t_{peak}$, $M_1$, $t_{99}$, and $L_{detect}$ were all much larger in the fixed reach studies (Table 4). These metrics all exhibited significant trends with discharge (Table 3), but the trends were not regularly consistent in their direction with the synoptic results. Overall, we found predominantly decreasing $t_{peak}$ with discharge in the fixed reaches - opposite to the synoptic finding - for 9 of 11 fixed reaches (and steeper Sen's slope in 9 of 11 fixed reaches). We also found decreasing $t_{99}$ with discharge in 9 of 11 fixed reaches (all with steeper Sen's slope than the synoptic), and decreasing $L_{detect}$ with discharge in 9 of 11 fixed reaches (all with steeper Sen's slope than the synoptic). Even with the longer reach lengths, relative to stream size, used

in the fixed reach studies, $L_{detect}$ averaged only ~2.0 m, and ranged from a maximum of 10 m to a minimum of 0.10 m.

With respect to short-term storage, we found increasing $M_I$ with increasing discharge in the synoptic
5   study, but this direction was reflected in only 2 of 11 fixed reaches. Sen's slope was larger in magnitude for 10 of the 11 fixed reaches, indicating $M_I$ interpreted from the fixed reach approach is more sensitive to discharge than the synoptic approach. We found overall decreasing $CV$, $\gamma$, and $H$ with increasing discharge in the synoptic study, indicating a decreasing importance of non-advective processes in the downstream direction along the network. The direction of this trend is consistent with 7 fixed reaches
10   for $CV$, 2 sites for $\gamma$, and 3 sites for $H$. Regardless of the direction of the relationship, the magnitude of Sens slope was larger for all fixed reaches compared to the synoptic study, indicating increased sensitivity to discharge relative to the synoptic sites.

For long-term storage and mass involved in advection-dispersion, we again found fixed-reach trends
15   were steeper and often opposed the direction of the trend for the synoptic data. For the synoptic study we found decreasing $f_{Qgainmax}$ (Fig. 4I) and $f_{Qlossmax}$ (Fig. 4L) with increasing discharge, which is consistent with 5 and 6 of the 11 fixed reaches, respectively. For the synoptic study we found an overall decreasing $f_{MAD}$ with increasing discharge, consistent with 7 of the 11 fixed reaches. The magnitude of Sens slope was larger for the fixed reaches than the synoptic study for $f_{MAD}$, $f_{Qgainmax}$, and $f_{Qlossmax}$.

The SAS analysis revealed decreasing sampling of the total storage zone ($f_{Vtot}$) with increasing discharge, but increasing $f_{Q,labeled}$ with discharge for the synoptic study. Together, these results indicate that increasing discharge in synoptic experiments resulted in sampling a larger fraction of the water exiting the reach, but smaller total volume of storage. Put another way, experiments in locations with
25   higher discharge were more likely to measure storage in (or proximal to) the stream channel at the expense of measuring more distal flowpaths and less-connected storage. For the fixed reach studies, we found decreasing $f_{Vtot}$ and $f_{Qlabeled}$ in 7 and 6 of the 11 reaches, respectively. In all cases, the magnitude of Sens slope was larger for the fixed reaches than the synoptic study.

Commented [KTV43]: This seems to be a new term. Perhaps it should be "$f_{Q,labelled}$"?

Commented [n44R43]: Wow! Thanks for your detailed attention. Yes, you are correct and has been corrected.

[revised manuscript text omitted]

Commented [KTV53]: Are the data used in this study available in accordance with the journal's data policy?

Commented [n54R53]: We added a statement here indicating that all data used in this analysis are made publicly available.

[revised manuscript text omitted]

**Page 3: [1] Deleted**          **Ward, Adam Scott**          **8/16/19 2:47:00 PM**

**Page 3: [1] Deleted**          **Ward, Adam Scott**          **8/16/19 2:47:00 PM**

**Page 3: [1] Deleted**          **Ward, Adam Scott**          **8/16/19 2:47:00 PM**

**Page 3: [1] Deleted**          **Ward, Adam Scott**          **8/16/19 2:47:00 PM**

**Page 3: [1] Deleted**          **Ward, Adam Scott**          **8/16/19 2:47:00 PM**

**Page 3: [1] Deleted**          **Ward, Adam Scott**          **8/16/19 2:47:00 PM**

**Page 3: [1] Deleted**          **Ward, Adam Scott**          **8/16/19 2:47:00 PM**

**Page 3: [1] Deleted**          **Ward, Adam Scott**          **8/16/19 2:47:00 PM**

**Page 3: [1] Deleted**          **Ward, Adam Scott**          **8/16/19 2:47:00 PM**

**Page 3: [1] Deleted**          **Ward, Adam Scott**          **8/16/19 2:47:00 PM**

**Page 3: [1] Deleted**          **Ward, Adam Scott**          **8/16/19 2:47:00 PM**

**Page 3: [2] Deleted**          **Ward, Adam Scott**          **8/17/19 1:12:00 PM**

**Page 3: [2] Deleted**          **Ward, Adam Scott**          **8/17/19 1:12:00 PM**

**Page 3: [2] Deleted**          **Ward, Adam Scott**          **8/17/19 1:12:00 PM**

**Page 3: [2] Deleted**          **Ward, Adam Scott**          **8/17/19 1:12:00 PM**

... [1]
... [2]
... [3]
... [4]
... [5]
... [6]
... [7]
... [8]
... [9]
... [10]
... [11]
... [12]
... [13]
... [14]
... [15]

| Page 3: [2] Deleted | Ward, Adam Scott | 8/17/19 1:12:00 PM |
| Page 3: [2] Deleted | Ward, Adam Scott | 8/17/19 1:12:00 PM |
| Page 3: [2] Deleted | Ward, Adam Scott | 8/17/19 1:12:00 PM |
| Page 3: [2] Deleted | Ward, Adam Scott | 8/17/19 1:12:00 PM |
| Page 3: [2] Deleted | Ward, Adam Scott | 8/17/19 1:12:00 PM |
| Page 3: [2] Deleted | Ward, Adam Scott | 8/17/19 1:12:00 PM |
| Page 3: [2] Deleted | Ward, Adam Scott | 8/17/19 1:12:00 PM |
| Page 3: [2] Deleted | Ward, Adam Scott | 8/17/19 1:12:00 PM |
| Page 3: [3] Deleted | Ward, Adam Scott | 8/16/19 2:47:00 PM |
| Page 3: [3] Deleted | Ward, Adam Scott | 8/16/19 2:47:00 PM |
| Page 3: [3] Deleted | Ward, Adam Scott | 8/16/19 2:47:00 PM |
| Page 3: [3] Deleted | Ward, Adam Scott | 8/16/19 2:47:00 PM |
| Page 3: [3] Deleted | Ward, Adam Scott | 8/16/19 2:47:00 PM |
| Page 3: [3] Deleted | Ward, Adam Scott | 8/16/19 2:47:00 PM |
| Page 3: [3] Deleted | Ward, Adam Scott | 8/16/19 2:47:00 PM |
| Page 3: [3] Deleted | Ward, Adam Scott | 8/16/19 2:47:00 PM |

... [20]
... [21]
... [22]
... [23]
... [24]
... [25]
... [26]
... [27]
... [28]
... [29]
... [30]
... [31]
... [32]
... [33]
... [34]

Yes, injections were instantaneous. Injection were not distributed across the channel. We selected appropriate mixing lengths to ensure the tracer was well mixed both laterally and vertically at the measurement sites.

While plunging is always a concern with using salt tracers, all locations were high-gradient, fast moving water with no deep, slowing moving pools, thus limiting potential concerns of plunging.

... [40]

| Page 12: [7] Deleted | Ward, Adam Scott | 8/17/19 12:38:00 PM |
|---|---|---|

| Page 13: [8] Commented [n24R23] | nschmadel | 8/1/19 4:58:00 PM |
|---|---|---|

Yes, we absolutely agree. Any storage flowpath that is longer in time than the window of detection is not measured with this tracer method. This limitation has led to us conducting the SAS analysis and offering best practices for tracer experiments later in the manuscript. We added "short-term" of "recovered" mass for clarity.

... [41]

... [42]

| Page 13: [9] Commented [KTV25] | King, Tyler Victor | 7/25/19 10:01:00 AM |
|---|---|---|

Variables were *italicized* in previous paragraphs. Suggest making formatting consistent throughout.

... [43]

| Page 13: [10] Commented [n26R25] | nschmadel | 8/1/19 4:56:00 PM |
|---|---|---|

Thanks again for your detailed attention. We will format all variables for consistency before this revision is resubmitted to the journal.

... [44]

| Page 13: [11] Commented [KTV27] | King, Tyler Victor | 7/25/19 9:58:00 AM |
|---|---|---|

Should this be $C_{ADE}$?
  Same with the second row in this equation. Otherwise, $C_{AD}$ would be a function of itself.

... [45]

| Page 13: [12] Commented [n28R27] | nschmadel | 8/1/19 4:57:00 PM |
|---|---|---|

You are correct. This has been corrected.

| Page 13: [13] Deleted | Ward, Adam Scott | 8/16/19 10:17:00 PM |
|---|---|---|

... [46]

| Page 28: [14] Formatted | Ward, Adam Scott | 8/16/19 10:59:00 PM |
|---|---|---|

Font: (Default) Times New Roman, Not Italic

... [47]

| Page 28: [14] Formatted | Ward, Adam Scott | 8/16/19 10:59:00 PM |
|---|---|---|

Font: (Default) Times New Roman, Not Italic

... [48]

| Page 28: [14] Formatted | Ward, Adam Scott | 8/16/19 10:59:00 PM |
|---|---|---|

Font: (Default) Times New Roman, Not Italic

... [49]

| Page 28: [14] Formatted | Ward, Adam Scott | 8/16/19 10:59:00 PM |
|---|---|---|

Font: (Default) Times New Roman, Not Italic

... [50]

| Page 28: [14] Formatted | Ward, Adam Scott | 8/16/19 10:59:00 PM |
|---|---|---|

Font: (Default) Times New Roman, Not Italic

... [51]

| Page 28: [14] Formatted | Ward, Adam Scott | 8/16/19 10:59:00 PM |
|---|---|---|

Font: (Default) Times New Roman, Not Italic

... [52]

... [53]

Font: (Default) Times New Roman, Not Italic

| Page 28: [15] Deleted | Ward, Adam Scott | 8/16/19 2:47:00 PM |

| Page 28: [15] Deleted | Ward, Adam Scott | 8/16/19 2:47:00 PM |

| Page 28: [16] Formatted | Ward, Adam Scott | 8/16/19 10:59:00 PM |

Font: (Default) Times New Roman, Not Italic

| Page 28: [16] Formatted | Ward, Adam Scott | 8/16/19 10:59:00 PM |

Font: (Default) Times New Roman, Not Italic

| Page 28: [17] Deleted | Ward, Adam Scott | 8/16/19 2:47:00 PM |

| Page 28: [17] Deleted | Ward, Adam Scott | 8/16/19 2:47:00 PM |

| Page 28: [18] Deleted | Ward, Adam Scott | 8/19/19 8:46:00 AM |

| Page 28: [18] Deleted | Ward, Adam Scott | 8/19/19 8:46:00 AM |

| Page 28: [19] Formatted | Ward, Adam Scott | 8/16/19 10:59:00 PM |

Font: (Default) Times New Roman, Not Italic

| Page 28: [19] Formatted | Ward, Adam Scott | 8/16/19 10:59:00 PM |

Font: (Default) Times New Roman, Not Italic

| Page 28: [20] Deleted | Ward, Adam Scott | 8/16/19 2:47:00 PM |

| Page 28: [20] Deleted | Ward, Adam Scott | 8/16/19 2:47:00 PM |

| Page 28: [20] Deleted | Ward, Adam Scott | 8/16/19 2:47:00 PM |

| Page 28: [21] Formatted | Ward, Adam Scott | 8/16/19 10:59:00 PM |

Font: (Default) Times New Roman, Not Italic

| Page 28: [21] Formatted | Ward, Adam Scott | 8/16/19 10:59:00 PM |

Font: (Default) Times New Roman, Not Italic

... [58]
... [59]
... [60]
... [61]
... [62]
... [63]
... [64]
... [65]
... [66]
... [67]
... [68]
... [69]
... [70]
... [71]
... [72]
... [73]
... [74]
... [75]
... [76]
... [77]
... [78]
... [79]

| Page 28: [23] Deleted | Ward, Adam Scott | 8/16/19 2:47:00 PM |

... [87]

| Page 28: [24] Deleted | Ward, Adam Scott | 8/19/19 8:47:00 AM |

... [88]

... [89]

| Page 28: [24] Deleted | Ward, Adam Scott | 8/19/19 8:47:00 AM |

... [90]

| Page 28: [25] Formatted | Ward, Adam Scott | 8/16/19 10:59:00 PM |

Font: (Default) Times New Roman, Not Italic

... [91]

... [92]

| Page 28: [25] Formatted | Ward, Adam Scott | 8/16/19 10:59:00 PM |

Font: (Default) Times New Roman, Not Italic

... [93]

| Page 28: [26] Deleted | Ward, Adam Scott | 8/16/19 2:47:00 PM |

... [94]

... [95]

... [96]

| Page 28: [26] Deleted | Ward, Adam Scott | 8/16/19 2:47:00 PM |

| Page 28: [27] Formatted | Ward, Adam Scott | 8/16/19 10:59:00 PM |

Font: (Default) Times New Roman, Not Italic

... [97]

... [98]

| Page 28: [27] Formatted | Ward, Adam Scott | 8/16/19 10:59:00 PM |

Font: (Default) Times New Roman, Not Italic

... [99]

| Page 28: [28] Deleted | Ward, Adam Scott | 8/16/19 2:47:00 PM |

... [100]

... [101]

... [102]

| Page 28: [28] Deleted | Ward, Adam Scott | 8/16/19 2:47:00 PM |

... [103]

| Page 28: [29] Formatted | Ward, Adam Scott | 8/16/19 10:59:00 PM |

... [104]

Font: (Default) Times New Roman, Not Italic

| Page 28: [29] Formatted | Ward, Adam Scott | 8/16/19 10:59:00 PM |

Font: (Default) Times New Roman, Not Italic

... [105]

... [106]

| Page 28: [30] Deleted | Ward, Adam Scott | 8/16/19 2:47:00 PM |

... [107]

... [108]

| Page 28: [30] Deleted | Ward, Adam Scott | 8/16/19 2:47:00 PM |

... [109]

... [110]

**Page 30: [33] Deleted**        **Ward, Adam Scott**        **8/16/19 11:04:00 PM**

... [117]

**Page 30: [33] Deleted**        **Ward, Adam Scott**        **8/16/19 11:04:00 PM**

... [118]

... [119]

**Page 30: [33] Deleted**        **Ward, Adam Scott**        **8/16/19 11:04:00 PM**

... [120]

**Page 30: [33] Deleted**        **Ward, Adam Scott**        **8/16/19 11:04:00 PM**

... [121]

... [122]

**Page 30: [33] Deleted**        **Ward, Adam Scott**        **8/16/19 11:04:00 PM**

... [123]

... [124]

**Page 30: [34] Formatted**        **Ward, Adam Scott**        **8/16/19 10:59:00 PM**

Font: (Default) Times New Roman, Not Italic

... [125]

**Page 30: [34] Formatted**        **Ward, Adam Scott**        **8/16/19 10:59:00 PM**

Font: (Default) Times New Roman, Not Italic

... [126]

**Page 30: [34] Formatted**        **Ward, Adam Scott**        **8/16/19 10:59:00 PM**

Font: (Default) Times New Roman, Not Italic

... [127]

... [128]

**Page 30: [34] Formatted**        **Ward, Adam Scott**        **8/16/19 10:59:00 PM**

Font: (Default) Times New Roman, Not Italic

... [129]

... [130]

**Page 30: [35] Deleted**        **Ward, Adam Scott**        **8/16/19 2:47:00 PM**

... [131]

... [132]

**Page 30: [35] Deleted**        **Ward, Adam Scott**        **8/16/19 2:47:00 PM**

... [133]

**Page 30: [36] Formatted**        **Ward, Adam Scott**        **8/16/19 10:59:00 PM**

... [134]

Font: (Default) Times New Roman, Not Italic

**Page 30: [36] Formatted**        **Ward, Adam Scott**        **8/16/19 10:59:00 PM**

Font: (Default) Times New Roman, Not Italic

... [135]

... [136]

**Page 30: [36] Formatted**        **Ward, Adam Scott**        **8/16/19 10:59:00 PM**

... [137]

Font: (Default) Times New Roman, Not Italic

... [138]

Font: (Default) Times New Roman, Not Italic

| Page 30: [38] Formatted | Ward, Adam Scott | 8/16/19 10:59:00 PM |
|---|---|---|

Font: (Default) Times New Roman, Not Italic

... [145]

| Page 30: [38] Formatted | Ward, Adam Scott | 8/16/19 10:59:00 PM |
|---|---|---|

Font: (Default) Times New Roman, Not Italic

... [146]

| Page 30: [38] Formatted | Ward, Adam Scott | 8/16/19 10:59:00 PM |
|---|---|---|

Font: (Default) Times New Roman, Not Italic

... [147]

| Page 30: [38] Formatted | Ward, Adam Scott | 8/16/19 10:59:00 PM |
|---|---|---|

Font: (Default) Times New Roman, Not Italic

... [148]

| Page 30: [38] Formatted | Ward, Adam Scott | 8/16/19 10:59:00 PM |
|---|---|---|

Font: (Default) Times New Roman, Not Italic

... [149]

| Page 31: [39] Formatted | Ward, Adam Scott | 8/16/19 10:59:00 PM |
|---|---|---|

Font: (Default) Times New Roman, Not Italic

... [150]

| Page 31: [39] Formatted | Ward, Adam Scott | 8/16/19 10:59:00 PM |
|---|---|---|

Font: (Default) Times New Roman, Not Italic

... [151]

| Page 31: [39] Formatted | Ward, Adam Scott | 8/16/19 10:59:00 PM |
|---|---|---|

Font: (Default) Times New Roman, Not Italic

... [152]

| Page 31: [40] Formatted | Ward, Adam Scott | 8/16/19 10:59:00 PM |
|---|---|---|

Font: (Default) Times New Roman, Not Italic

... [153]

| Page 31: [40] Formatted | Ward, Adam Scott | 8/16/19 10:59:00 PM |
|---|---|---|

Font: (Default) Times New Roman, Not Italic

... [154]

| Page 31: [40] Formatted | Ward, Adam Scott | 8/16/19 10:59:00 PM |
|---|---|---|

Font: (Default) Times New Roman, Not Italic

... [155]

| Page 31: [40] Formatted | Ward, Adam Scott | 8/16/19 10:59:00 PM |
|---|---|---|

Font: (Default) Times New Roman, Not Italic

... [156]

| Page 31: [41] Deleted | Ward, Adam Scott | 8/16/19 2:47:00 PM |
|---|---|---|

... [157]

| Page 31: [41] Deleted | Ward, Adam Scott | 8/16/19 2:47:00 PM |
|---|---|---|

... [158]

| Page 31: [42] Formatted | Ward, Adam Scott | 8/16/19 10:59:00 PM |
|---|---|---|

... [159]

Font: (Default) Times New Roman, Not Italic

... [160]

**Page 31: [43] Deleted**              **Ward, Adam Scott**              **8/16/19 2:47:00 PM**

**Page 31: [44] Formatted**              **Ward, Adam Scott**              **8/16/19 10:59:00 PM**

Font: (Default) Times New Roman, Not Italic

**Page 31: [44] Formatted**              **Ward, Adam Scott**              **8/16/19 10:59:00 PM**

Font: (Default) Times New Roman, Not Italic

**Page 31: [44] Formatted**              **Ward, Adam Scott**              **8/16/19 10:59:00 PM**

Font: (Default) Times New Roman, Not Italic

**Page 31: [44] Formatted**              **Ward, Adam Scott**              **8/16/19 10:59:00 PM**

Font: (Default) Times New Roman, Not Italic

**Page 31: [44] Formatted**              **Ward, Adam Scott**              **8/16/19 10:59:00 PM**

Font: (Default) Times New Roman, Not Italic

**Page 31: [44] Formatted**              **Ward, Adam Scott**              **8/16/19 10:59:00 PM**

Font: (Default) Times New Roman, Not Italic

**Page 31: [45] Deleted**              **Ward, Adam Scott**              **8/16/19 2:47:00 PM**

**Page 31: [45] Deleted**              **Ward, Adam Scott**              **8/16/19 2:47:00 PM**

**Page 31: [46] Formatted**              **Ward, Adam Scott**              **8/16/19 10:59:00 PM**

Font: (Default) Times New Roman, Not Italic

**Page 31: [46] Formatted**              **Ward, Adam Scott**              **8/16/19 10:59:00 PM**

Font: (Default) Times New Roman, Not Italic

**Page 31: [46] Formatted**              **Ward, Adam Scott**              **8/16/19 10:59:00 PM**

Font: (Default) Times New Roman, Not Italic

**Page 31: [46] Formatted**              **Ward, Adam Scott**              **8/16/19 10:59:00 PM**

Font: (Default) Times New Roman, Not Italic

**Page 31: [46] Formatted**              **Ward, Adam Scott**              **8/16/19 10:59:00 PM**

Font: (Default) Times New Roman, Not Italic

**Page 31: [46] Formatted**              **Ward, Adam Scott**              **8/16/19 10:59:00 PM**

Font: (Default) Times New Roman, Not Italic

Font: (Default) Times New Roman, Not Italic

| Page 31: [46] Formatted | Ward, Adam Scott | 8/16/19 10:59:00 PM |
|---|---|---|

Font: (Default) Times New Roman, Not Italic

| Page 31: [46] Formatted | Ward, Adam Scott | 8/16/19 10:59:00 PM |
|---|---|---|

Font: (Default) Times New Roman, Not Italic

| Page 31: [47] Formatted | Ward, Adam Scott | 8/16/19 10:59:00 PM |
|---|---|---|

Font: (Default) Times New Roman, Not Italic

| Page 31: [47] Formatted | Ward, Adam Scott | 8/16/19 10:59:00 PM |
|---|---|---|

Font: (Default) Times New Roman, Not Italic

| Page 31: [48] Deleted | Ward, Adam Scott | 8/16/19 2:47:00 PM |
|---|---|---|

| Page 31: [48] Deleted | Ward, Adam Scott | 8/16/19 2:47:00 PM |
|---|---|---|

| Page 31: [49] Formatted | Ward, Adam Scott | 8/16/19 10:59:00 PM |
|---|---|---|

Font: (Default) Times New Roman, Not Italic

| Page 31: [49] Formatted | Ward, Adam Scott | 8/16/19 10:59:00 PM |
|---|---|---|

Font: (Default) Times New Roman, Not Italic

| Page 32: [50] Formatted | Ward, Adam Scott | 8/16/19 10:59:00 PM |
|---|---|---|

Font: (Default) Times New Roman, Not Italic

| Page 32: [51] Formatted | Ward, Adam Scott | 8/16/19 10:59:00 PM |
|---|---|---|

Font: (Default) Times New Roman

| Page 32: [52] Formatted | Ward, Adam Scott | 8/16/19 10:59:00 PM |
|---|---|---|

Font: (Default) Times New Roman, Not Italic

| Page 32: [53] Formatted | Ward, Adam Scott | 8/16/19 10:59:00 PM |
|---|---|---|

Font: (Default) Times New Roman

| Page 32: [54] Formatted | Ward, Adam Scott | 8/16/19 10:59:00 PM |
|---|---|---|

Font: (Default) Times New Roman

| Page 32: [55] Formatted | Ward, Adam Scott | 8/16/19 10:59:00 PM |
|---|---|---|

Font: (Default) Times New Roman

| Page 32: [56] Formatted | Ward, Adam Scott | 8/16/19 10:59:00 PM |
|---|---|---|

Font: (Default) Times New Roman, Not Italic

... [189]
... [190]
... [191]
... [192]
... [193]
... [194]
... [195]
... [196]
... [197]
... [198]
... [199]
... [200]
... [201]
... [202]
... [203]
... [204]

Font: (Default) Times New Roman, Not Italic

| Page 32: [61] Formatted | Ward, Adam Scott | 8/16/19 10:59:00 PM |
|---|---|---|

Font: (Default) Times New Roman

| Page 32: [62] Formatted | Ward, Adam Scott | 8/16/19 10:59:00 PM |
|---|---|---|

Font: (Default) Times New Roman

| Page 32: [63] Formatted | Ward, Adam Scott | 8/16/19 10:59:00 PM |
|---|---|---|

Font: (Default) Times New Roman, Not Italic

| Page 32: [64] Formatted | Ward, Adam Scott | 8/16/19 10:59:00 PM |
|---|---|---|

Font: (Default) Times New Roman

| Page 32: [65] Formatted | Ward, Adam Scott | 8/16/19 10:59:00 PM |
|---|---|---|

Font: (Default) Times New Roman, Not Italic

| Page 32: [66] Formatted | Ward, Adam Scott | 8/16/19 10:59:00 PM |
|---|---|---|

Font: (Default) Times New Roman

| Page 32: [67] Formatted | Ward, Adam Scott | 8/16/19 10:59:00 PM |
|---|---|---|

Font: (Default) Times New Roman

| Page 32: [68] Deleted | Ward, Adam Scott | 8/16/19 10:59:00 PM |
|---|---|---|

| Page 32: [69] Formatted | Ward, Adam Scott | 8/16/19 10:59:00 PM |
|---|---|---|

Font: (Default) Times New Roman, Highlight

| Page 32: [70] Formatted | Ward, Adam Scott | 8/16/19 10:59:00 PM |
|---|---|---|

Font: (Default) Times New Roman, Not Italic, Highlight

| Page 32: [71] Formatted | Ward, Adam Scott | 8/16/19 10:59:00 PM |
|---|---|---|

Font: (Default) Times New Roman, Highlight

| Page 32: [72] Formatted | Ward, Adam Scott | 8/16/19 10:59:00 PM |
|---|---|---|

Font: (Default) Times New Roman, Not Italic, Highlight

| Page 32: [73] Formatted | Ward, Adam Scott | 8/16/19 10:59:00 PM |
|---|---|---|

Font: (Default) Times New Roman, Highlight

| Page 32: [74] Formatted | Ward, Adam Scott | 8/16/19 10:59:00 PM |
|---|---|---|

Highlight

| Page 32: [75] Formatted | Ward, Adam Scott | 8/16/19 10:59:00 PM |
|---|---|---|

Font: (Default) Times New Roman, Highlight

| Page 32: [76] Formatted | Ward, Adam Scott | 8/16/19 10:59:00 PM |
|---|---|---|

... [238]

... [239]

... [240]

... [241]

... [242]

---

## Author Response (AR2)

Response to comments from Anonymous Referee #2

Ward et al., have compiled a uniquely thorough dataset and compiled a number of analyses to better explain the effect of river corridor exchange across a range of scales up to the river network scale and specifically to test Wondzell's conceptual model. The work shows systematic changes in a variety of metrics moving downstream through the watershed. Interestingly, the study found an increase in K in the downstream direction, which defied initial expectations. The work also verified that discharge is an inappropriate scaling metric for prediction of river corridor exchange because it is dependent on a variety of forcing mechanisms operating across a range of scales. This paper will be a significant contribution to the field once it is in print.

The authors have made revisions to the manuscript based on comments from a fellow reviewer and myself. I am generally happy with their responses and how the edits addressed the comments, but have a few remaining concerns--most of which are minor. I think they would all help improve the paper and broaden its accessibility for those who are not engaged in this immediate field.

> No modification required based on the comments above.

My largest remaining concern is the justification for the large number of authors included on this paper. Most top-tier journals (e.g. Science and Nature, as well as many others) have begun to require a thorough justification for the inclusion of each author that includes some description of how each author contributed. However this goal is met, I think it worthwhile to have something more descriptive than the current three-line 'author contribution' section, which only mentions two of the 37 authors. Generally, my understanding of authorship is that each author has contributed significantly to at least two of the four tiers of authorship – funding, writing, conception, work. I believe the authors that they have all met this requirement, but I think it is generally worthwhile to explicitly describe the ways in which each contributed, especially when so many authors are included and when the alphabetical listing confuses the amount of contribution that each made. I previously suggested the model of Clement (2014) to show how justification could be demonstrated, but additional models exist. If length is a concern, justification could be included in the supplement.

> We have added a detailed listing of contributions to the manuscript using the Contributor Role Taxonomy (CRediT) scheme.

Minor comments:
Page:line
8:40 I appreciate how the authors presented their justification for basing field-estimation of mixing lengths based on experience, rather than an empirical estimate. I think it would be helpful to report the mixing lengths that were actually used for each of the streams, perhaps only as a scatter plot showing q vs mixing length. This is a helpful metric to show to readers less-familiar with this field.

> Mixing lengths were not recorded in the field. While we agree this could be informative, we do not have the data to provide this visualization.

10-11:section 2.2. This section is very helpful for putting the authors goals and the limitations of this approach in perspective.

Thank you. No modifications in response to this comment.

12:1 Table is cut-off and I could not evaluate fully, but I think this is a worthwhile addition in response to one of the reviewer comments. This summary of "key metrics" and approaches helps with clarity of purpose. The included subset of abbreviations are helpful for the reader to decipher the large number of terms included in this paper. That said, I still find it difficult to wade through the large number of terms included in the equations of this paper and think a single table of variable definitions would be very useful--similar to that included in engineering-focused journals. Someone more familiar with the field will find this less-difficult than I did, most likely, however, the task of sorting through so many equations across so many pages of text to identify the meaning of each term and equation is a barrier for entry for those less familiar with this topic.

We have added a table including symbols, definitions, and units as a new section following the conclusions of the study.

14:20 I think it might be easier to follow notation if the normalized breakthrough curve (concentration) notation were also capitalized ('c' --> 'C') as the other concentration notations are.

We use the lower case "c" to differentiae the normalized concentration from the observed concentration ("C"). The alternative here would be to replace "c" with "$C_{norm}$" at all instances, which we believe is more confusing to the reader. We have elected not to modify the text in response to this point.

20:20 I wonder why the references to panels in figure 3 are out-of-order.

We have reorganized the paragraph in question to present the panels of Fig. 3 in order.

23:20/Figure 4 It would be helpful to include r2 values on the plots, especially since they're referenced here.

We have added $r^2$ to each panel of the figure as suggested.

27:Figure 5 I still disagree that adding lines of best fit on this plot is useful or appropriate. I think they imply trends that I do not see in the data—especially in panel A. I think the authors should at least include an r2 value to emphasize the poor fit.

We have added $r^2$ to each panel of the figure as suggested.

28:28, a reference back to figure 3 would be helpful here

Reference to Fig. 3D added as suggested.

27:7 & 30:Fig. 6. I think it would be helpful to walk the reader through panels b-f of figure 6. I'm not sure the points that are being made in the figure besides that there's some goodness of fit.

We have added a sentence to section 4.1 to further explain these panels.